# Ultrasound-induced reorientation for multi-angle optical coherence tomography

Mia Kvåle Løvmo [1,3], Shiyu Deng [2,3], Simon Moser [1,3], Rainer Leitgeb [2], Wolfgang Drexler[2] & Monika Ritsch-Marte [1] ✉

Organoid and spheroid technology provide valuable insights into developmental biology and oncology. Optical coherence tomography (OCT) is a label-free technique that has emerged as an excellent tool for monitoring the structure and function of these samples. However, mature organoids are often too opaque for OCT. Access to multi-angle views is highly desirable to overcome this limitation, preferably with non-contact sample handling. To fulfil these requirements, we present an ultrasound-induced reorientation method for multi-angle-OCT, which employs a 3D-printed acoustic trap inserted into an OCT imaging system, to levitate and reorient zebrafish larvae and tumor spheroids in a controlled and reproducible manner. A model-based algorithm was developed for the physically consistent fusion of multi-angle data from a priori unknown angles. We demonstrate enhanced penetration depth in the joint 3D-recovery of reflectivity, attenuation, refractive index, and position registration for zebrafish larvae, creating an enabling tool for future applications in volumetric imaging.

A steep increase in organoid and spheroid research could be witnessed in recent years, providing vital insights into developmental biology and oncology. A strong motivation for this is the potential of organoids and cancer spheroids to reduce animal experimentation to some extent[1]. Spheroids can be grown with the support of an extracellular matrix or scaffold-free[2]. Organoids are usually grown in Matrigel, derived from the secretion of a type of mouse sarcoma cells. Matrigel is complex and variable which gives rise to a certain irreproducibility. Moreover, it is known that the matrix scaffolds have a mechanical impact that is often poorly understood or characterized[3]. Therefore, in the past few years, considerable effort has been directed towards Matrigel-free organoid growth[4].

In response to such problems, contact-free levitation of samples has been sought. On the single-cell level, it is possible to use holographic optical tweezers to induce rotations for tomographic studies[5,6], but for cell clusters reaching mm-size, optical trapping would lead to over-heating due to the intensities needed to counteract

the growing weight[7], even in the most favorable case of counter-propagating optical beams creating a trapping region between two laser spots, as in the macro-tweezers system[8]. In this context, please note that the limits of optical trapping stated in ref. 9 are incorrect, significantly overestimating heating under normal operating conditions. Nevertheless, to tackle biological samples beyond what optical tweezers can achieve, ultrasound techniques for levitation and actuated handling were developed: Standing bulk acoustic waves (BAW) operating at (sub-)MHz frequencies push biological cells into low-pressure regions (planes, lines, or spots depending on the number of orthogonal standing waves)[9–11], and by modulation of the acoustic waves, objects can be transiently or continuously rotated[12–16]. Surface acoustic waves (SAW) can also be used, e.g., to create acoustofluidic rotational tweezers for morphological phenotyping of zebrafish larvae[17]. Streaming vortices generated by acoustically induced oscillations of bubbles or solid structures have been applied to rotate zebrafish embryos[18], single cells, pollen grains, and nematodes[19–22]. For

[1]Institute of Biomedical Physics, Medical University of Innsbruck, Innsbruck, Austria. [2]Center for Medical Physics and Biomedical Engineering, Medical University of Vienna, Vienna, Austria. [3]These authors contributed equally: Mia Kvåle Løvmo, Shiyu Deng, Simon Moser. ✉e-mail: monika.ritsch-marte@i-med.ac.at

compatibility with scanning-based imaging modalities, however, BAW operation is favorable, as it supports tilting the sample into various stationary (non-rotational) orientations. Bulk waves also simplify device up-scaling for the manipulation of large samples far from the chamber boundaries. Scaffold-free confinement in the center of the chamber represents a big advantage of acoustic levitation since it makes the system more open to performing various assays, such as irradiating parts of the sample by light, adding chemicals and pharmaceuticals by micro-fluidics, or mechanical probing with tips.

Optical coherence tomography (OCT)[23–25], a technique based on low-coherence interferometry, can reconstruct micrometer sample morphology from the backscattered light with high imaging speed (MHz A-scan rate) and has been widely employed for (bio)medical applications. OCM, which uses a higher numerical aperture (NA), can achieve high lateral resolution but usually is limited in penetration depth. OCT/OCM has emerged as a valuable imaging modality for living tissues and model organisms[26–28] and more recently for organoids and spheroids[29–31], providing high-resolution information on the internal structural organization inside the organoids non-invasively and label-free.

Nevertheless, mature specimens often become optically dense, and intractable not only for optical diffraction tomography (ODT)[32,33] but also for OCT, leading to shadows and limited tissue morphology information. Shadow removal algorithms have been developed for OCT images of the optic nerve head[34–36]. However, removing the shadows cast by high-attenuation structures like the eye of a wild-type zebrafish larva remains challenging, because the OCT incident light can be fully occluded. 3D optical coherence refraction tomography[37] compensates for this issue by controlling the incident beam angle and position using a parabolic mirror, but this was limited to ±75° angular orientation and needed to immobilize samples like zebrafish embryos in agarose gel.

In this work, we present an easy-to-use solution overcoming the above-explained limitations and problems encountered when imaging organoids, spheroids, or developing organisms: ULTrasound-Induced reorientation for Multi-Angle OCT (ULTIMA OCT) uses a small add-on microfluidic chamber with tunable BAW to stably and reproducibly rotate the sample into several orientations. This enables OCT imaging from different viewing angles in the full range of 360° around the sample's major axis, which makes 3D tomographic reconstruction feasible also for optically dense samples. The immobilization of the sample is contact-free, and neither involves any rotating mechanical parts, nor any elements obstructing optical imaging or introducing optical aberrations.

The price to pay in this approach is the fact that one cannot precisely choose the exact viewing angles, since the stable trapping positions and orientations in the acoustic force fields to some extent also depend on the unknown sample itself[11]. However, we provide a generally applicable solution, i.e., a model-based algorithm, which can deal with the added complexity of tomographic reconstruction with viewing angles that are not precisely known a priori.

## Results

### Working principle of ULTIMA-OCT

The workflow of ULTIMA-OCT and its ingredients are explained schematically in Fig. 1, and a schematic animation of the data acquisition procedure can be found in Supplementary Movie 1. Acoustic radiation forces are used to levitate the sample and to induce transient rotations in a fluid chamber. Each new orientation is a stable trapping position, and we perform OCM scanning of the sample at a desired number of orientation angles. The 3D OCM data is then processed by a model-based algorithm to extract the underlying reflectivity map, while also yielding information about the attenuation and the refractive index (RI) contrast maps as well as the position and orientation parameters of the trapped sample.

### Acoustic actuation

The acoustic manipulation chamber consists of a 3D printed frame with a symmetric octagonal cross-section with four piezo-electric plate transducers and four reflectors around its sides (Fig. 2 and Supplementary Fig. 1). We apply a sinusoidal signal to the transducers to propagate BAWs in four directions in the liquid-filled chamber and generate acoustic standing waves in each direction upon reflection. The standing waves have pressure nodes every $\lambda/2$ ($\approx 1\,mm$ around 600 kHz) along each propagation direction in the fluid chamber. The front and back of the octagon frame are sealed with a coverslip and an aluminum plate, respectively, and imaging is performed through the bottom coverslip that also acts as the reflector for the acoustic waves from the top transducer. Our current device is developed to accommodate a range of sample sizes and shapes, and we have demonstrated this by manipulating samples, from highly asymmetric mm-sized 3–5 days post fertilization (dpf) zebrafish embryos to less asymmetrically shaped sub-mm-sized melanoma spheroids. We rely on the resonant enhancement of the waves in the fluid chamber to get sufficient force to levitate our targeted biological samples. Levitation by the top transducer was achieved at a minimum peak-to-peak driving voltage of 20 V, corresponding to a maximum pressure amplitude of 80 kPa (see "Methods" and Supplementary Methods for details on used chamber dimensions and acoustic resonances).

To characterize and optimize our contact-less trapping platform for reorientation and multi-angle image acquisition, we used fixated 3 dpf zebrafish embryos, as they are readily available samples that are perfectly suited to demonstrate the benefits of our approach. To observe the zebrafish while tuning the acoustic settings for stable reorientation, we used an inverted microscope with oblique illumination through the front cover glass of the chamber, acquiring dark-field images (Fig. 2c and details in Supplementary Methods). With the acoustic radiation forces[38–41] from the top transducer, we levitate the sample against gravity, in one of the nodal planes in the center region of the chamber where all four acoustic waves intersect. With the additional radiation forces from one side transducer, we align the sample with its major axis to the length of the chamber ($y$-axis in Fig. 2).

By changing the voltage on the transducers, and hence the relative magnitudes of the acoustic radiation forces in each direction, we generate an acoustic restoring torque[11,42,43] acting on the sample. The torque direction is perpendicular to the acoustic propagation directions, hence parallel to the $y$-axis, and the direction of rotation is in the $xz$-plane (Fig. 2). With a dominating top transducer, the sample is aligned with its minor axis to the steepest trap-stiffness in $z$-direction (90° in Fig. 2c). When we increase the amplitude of one of the side transducers step by step, we rotate the pressure landscape and the sample is rotated in a step-wise fashion until the sample is aligned with its minor axis to the now dominating forces from the side transducer. We alternate between increasing and decreasing the amplitudes between pairs of transducers in a sequence, to rotate the sample 360° about its major axis, while ensuring levitation (top transducer voltage is tuned, but never zero).

Moreover, we found that for sufficiently asymmetric samples one can precisely control the orientation in a more efficient way, by exciting two overlapping orthogonal modes at exactly the same frequency in the chamber (by e.g., the top transducer and the orthogonal side transducer S3 in Fig. 2) and adjusting the relative amplitude and phase (see Supplementary Methods for details on acoustic actuation), similar to ref. 12 but with additional levitation in our upright (not horizontal) chamber. In Fig. 2c, we show dark-field images (two stitched tiles per image) from only eight different orientations of a 3 dpf zebrafish, but we can reorient this sample with a much finer step-size, see Supplementary Movie 2, showing reorientation by two transducers of a zebrafish embryo.

To demonstrate our capability of extending the outlined acoustic manipulation to other types of samples, we also trapped and

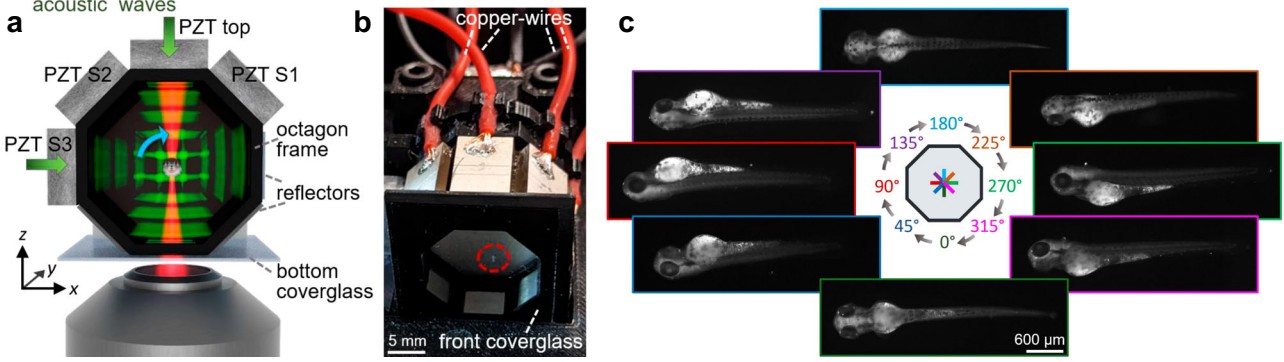

**Fig. 1 | Schematics and workflow of ULTrasound-Induced reorientation for Multi-Angle-OCT (ULTIMA-OCT). a** Depicts the fluid-filled acoustic chamber in which the sample is levitated and reoriented by means of acoustic actuation. The specimen is rotated in a step-wise manner into several stable trapping positions (**b**), and optical coherence microscopy (OCM) imaging is performed in each of them (**c**). The acquired OCM data is post-processed (**d**) and fed into a multiscale optimization algorithm (**e**), which performs a fusion of the images and outputs 3D reconstructions (**f**) of reflectivity $R$, attenuation $\alpha$, and refractive index (RI) contrast $\Delta n$. In (**g**), an exemplary collection of samples is shown, where ULTIMA-OCT can be applied.

**Fig. 2 | Acoustic actuation for multi-angle imaging. a** An illustration of acoustic manipulation of levitated zebrafish embryo (not to scale). By coupling bulk acoustic waves into the fluid-filled chamber from multiple directions, acoustic standing waves (green) are generated upon reflection, to levitate the sample and induce transient rotations for optical imaging (red beam), e.g., for multi-angle high-speed OCM through the bottom cover glass of the 3D printed octagon frame (black). The direction of rotation in the *xz*-planes is indicated (blue arrow). **b** Assembled octagon chamber with levitated zebrafish embryo (inside stippled red circle), scale bar: 5 mm. **c** The optimization of the acoustic actuation can be carried out on an inverted microscope with optical image acquisition. As an example, darkfield (oblique illumination) images of a wild-type 3 dpf zebrafish embryo are shown here, for a selection of eight chosen angles of acoustic reorientation, scale bar: 600 µm.

reoriented melanoma spheroids (see Supplementary Fig. 4). These samples were smaller and less asymmetric than the zebrafish embryos, but we could reorient them around their major axis by the same-frequency two-transducer actuation described above. For other settings, however, sustained rotation was induced, with a rotation direction that could be reversed by changing the relative phase between the two orthogonal transducers by 180°, as also demonstrated by others[13] (see Supplementary Movie 3). It has so far always been possible to avoid unwanted sustained rotation and to just stably reorient the spheroid 360° around one axis. In each orientation the sample is held stably without any significant motion, see Supplementary Movie 2 and 3. Our trapping platform can accommodate a large range

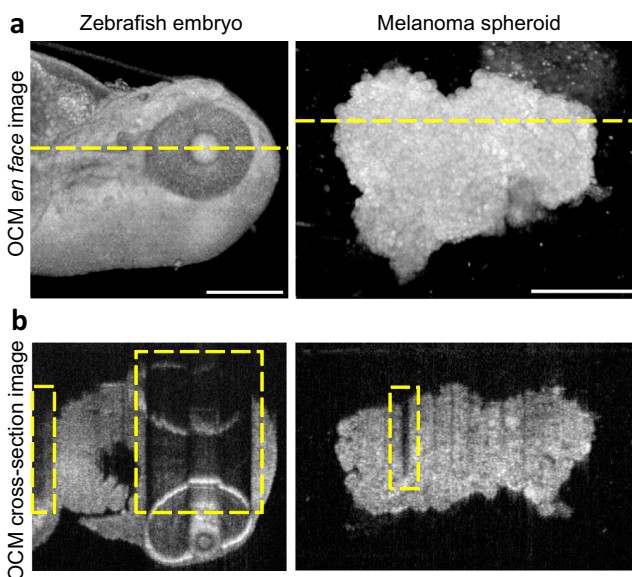

**Fig. 3 | OCM limitations. a** An average intensity projection for zebrafish embryo and standard deviation projection for melanoma spheroid, both in logarithmic scale (OCM en face images). **b** Cross-section images for zebrafish embryo and melanoma spheroid in logarithmic scale. Cross-section image positions are indicated by the yellow dash lines in the OCM en face images in (**a**). Shadow artifacts are indicated by the yellow dashed boxes in the OCM cross-section images. Scale bars: 200 μm.

of sample sizes and shapes, but each new sample needs its own fine-tuning of the acoustic settings, which is performed on the fly while directly observing the object.

## High-resolution OCT imaging

To verify the feasibility of ULTIMA-OCT, biological samples such as fixated zebrafish larvae and melanoma spheroids were tested. The OCM system successfully captured the 3D data of the samples levitated and reoriented in the acoustic chamber. Figure 3 shows OCM images of a 3 dpf less-pigmented *Mitfa*$^{b692/b692}$/*ednrb1*$^{b140/b140}$ zebrafish and a melanoma spheroid, respectively, imaged from one direction.

The cross-section images (locations are indicated by the yellow dashed lines in the en face images in Fig. 3a) exhibit distinct shadow artifacts, as seen in Fig. 3b. For the zebrafish embryo, the eye with high melanin content and the yolk with high-attenuating internal structures cast shadows (signal loss) on deeper morphological features (marked by yellow dashed boxes in the OCM cross-section images). Similar artifacts were also identified in melanoma spheroids, where cells with high melanin levels limited the penetration depth due to high absorption. These shadow artifacts from a single viewing angle were also clearly revealed in the 3D rendering (see Supplementary Movie 4. Note that the structure below the sample is not an artifact, but an accidental contamination.).

En face images obtained by average intensity projection or standard deviation projection display the combined signal from different sample depths. Naturally, the artifacts caused by shadowing are not as obvious in this type of visualization. Figure 4 demonstrates such OCM images, with less noticeable shadowing, of a 5 dpf less-pigmented zebrafish embryo obtained from eight viewing angles. Whole-body en face data were obtained from three angles (indicated by the sub-image frame colors). Zebrafish features such as eye, otolith, yolk, muscle, notochord, and fins were discerned clearly by the OCM setup. Complementary zebrafish features were visualized from individual angles, but darker regions were observed in the images, depending on the reorientation angles.

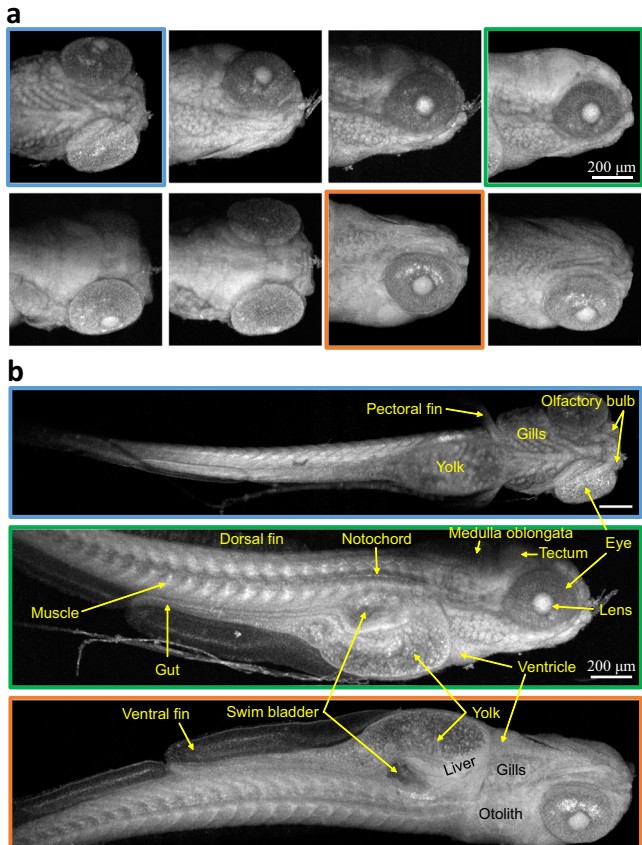

**Fig. 4 | OCM en face images of acoustically reoriented 5 dpf zebrafish embryo. a** OCM en face images of the head section and **b** full body images of a zebrafish embryo (mutation *Mitfa*$^{b692/b692}$/*ednrb1*$^{b140/b140}$). Frame colors indicate corresponding angles. Scale bars: 200 μm.

## Reconstruction

The sample can be reoriented in a reproducible manner by acoustic actuation, but in contrast to externally induced mechanical rotation of an object immobilized in a container, the exact orientation between the recorded OCM volumes is unknown a priori. Depending on the orientation, different parts of the sample are occluded due to attenuation by structures in the sample, which for a zebrafish embryo is especially pronounced for the eye and the yolk sac. Also, structures of different RI inside the sample cause a local delay or surge of the recorded A-scan[37,44,45]. This effect is especially visible as a delay for structures behind the lens portion of the eye.

Due to the mentioned distortion, the OCM volumes belonging to different orientation angles are not simply related by a rigid body transform, but correspond to each other in a more complicated manner. Moreover, the shadowing artifacts hinder a reliable registration of the orientations of the different volumes. Therefore, we formulate the fusion as an inverse problem, where the OCM image formation is expressed as a physical forward model. This approach grants us the flexibility to deal with these uncertainties by constraints and regularization. This includes total variation (TV) and Tikhonov regularization, as well as positivity and object support constraints. We solve this inverse problem by means of a gradient-based optimization approach, whose optimization parameters consist in the underlying reflectivity map $R$ as well as attenuation $\alpha$, RI contrast $\Delta n$, and motion parameters $q$ (rotation parameterized by unit quaternions) and $t$ (translation). For a detailed description, we refer to the Supplementary Methods.

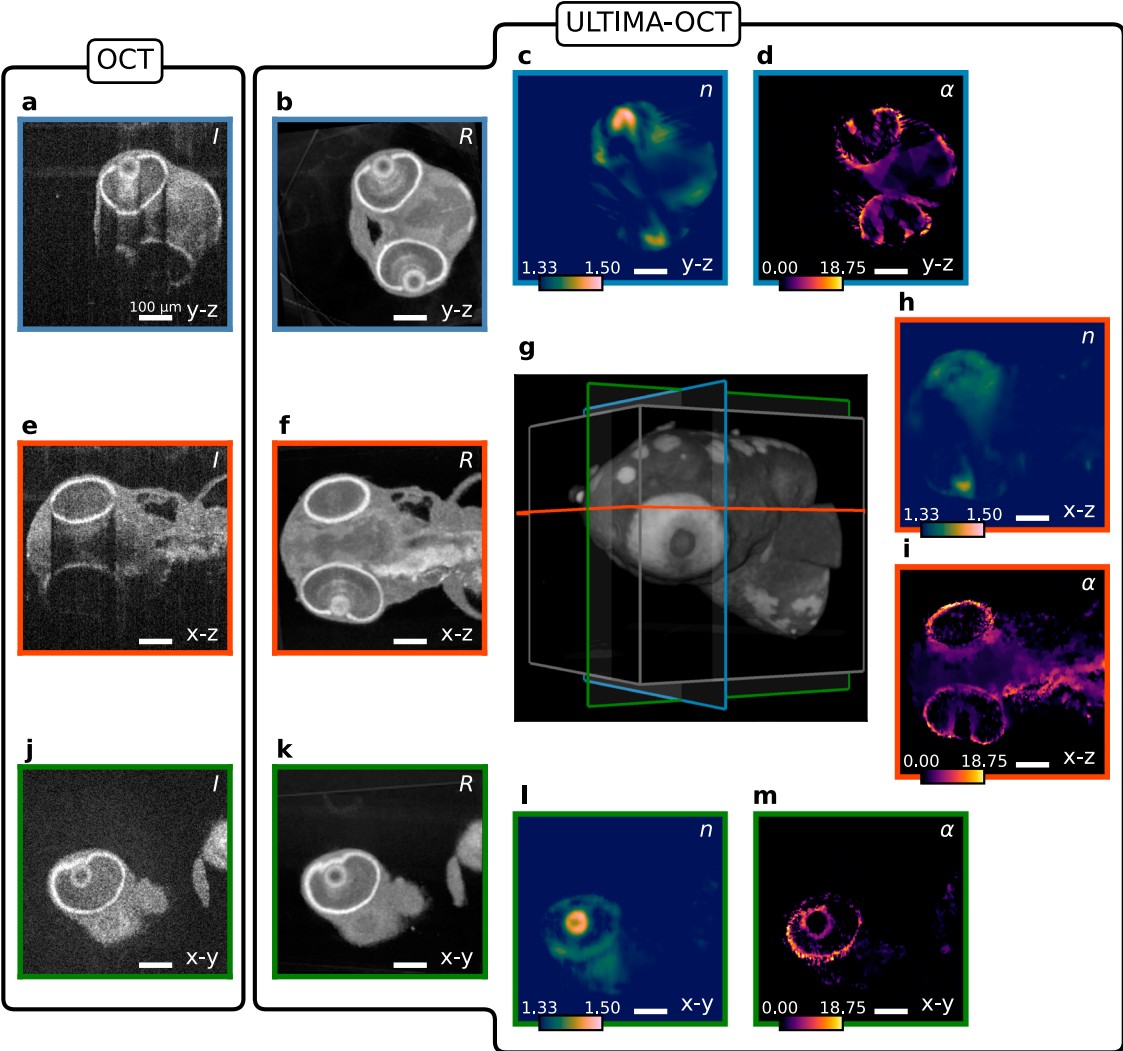

**Fig. 5 | Reconstruction results for the head section of a 3 dpf wild-type zebrafish embryo. a–d** The (*y–z*), **e**, **f**, **h**, **i** the (*x–z*), and **j–m** depict the (*x–y*) sections of the reconstructions. The leftmost column (**a**, **e**, **j**) shows the sections of the recorded OCM volumes in logarithmic scale, whereas the adjacent column (**b**, **f**, **k**) shows the reconstructed reflectivity map *R* of the same sections in logarithmic scale.

**d**, **i**, **m** Show the slices of the reconstructed attenuation map *α* (in mm⁻¹), whereas **c**, **h**, **l** depict the sections through the reconstructed RI distribution *n*. In (**g**), a 3D rendering of the reconstructed reflectivity map is shown, together with the planes shown in (**a–f**) and (**h–m**). Scale bars: 100 μm.

Figures 5 and 6 show the reconstruction results of ULTIMA-OCT from the head section of 3 dpf zebrafish embryos, a wild-type and a less-pigmented mutation, respectively. In a comparison of the OCM volumes and the reconstruction of the reflectivity map *R*, both depicted in logarithmic scale, one clearly appreciates the benefit of the proposed approach. One can see that both specimens strongly attenuate the signal in the OCM volumes, whereas the reconstructed reflectivities no longer show attenuation or distortion artifacts. In the wild-type zebrafish, the scattering and absorptive structures contained within the total attenuation *α* are present across the whole head section, whereas for the mutated specimen the eyes account for most of the attenuation. Vertebrate eye lenses exhibit a graded-index (GRIN) profile, which increases towards the center. As the most prominent structures of RI map are those belonging to the eye lenses, we employ regularization that promotes smoothness, whereas, for the attenuation and reflectivity maps, we use edge-preserving regularization. The values obtained for the RI using this method are consistent with values from the literature⁴⁶,⁴⁷. The dataset of the wild-type zebrafish embryo consists of OCM recordings of 11 angles, which are roughly distributed between 0 and 360°, whereas for the less pigmented mutation zebrafish embryo, ten angles were used. 3D visualization of reconstructions

comparing single- and multi-view of the wild-type zebrafish embryo can be found in Supplementary Movie 5 an 6 with a 3D rendered object and a flythrough-animation, respectively.

## Discussion

ULTIMA-OCT imaging and tomographic reconstruction provide volumetric information on the sample with enhanced penetration depth. This is achieved by acoustic reorientation, without the need for any moving mechanical parts, and in a non-contact way without the need of a supporting scaffold. This makes the sample much more accessible for mechanical probing and facilitates unobstructed and undistorted imaging. We will now discuss some current difficulties and limitations as well as possible extensions of the approach in the future.

Ultrasound trapping offers several advantages for OCT imaging. The standard method to study samples such as live zebrafish larvae with OCT has been to manually position the larvae under anesthesia on a gel layer on a coverslip and image from one direction. The purpose of this gel layer is to lift the sample up to create sufficient distance from the coverslip, and the coverslip is usually tilted to get rid of the reflection from the bottom coverslip. As a result, parts of the big sample may be out of focus and require focus stacking or image

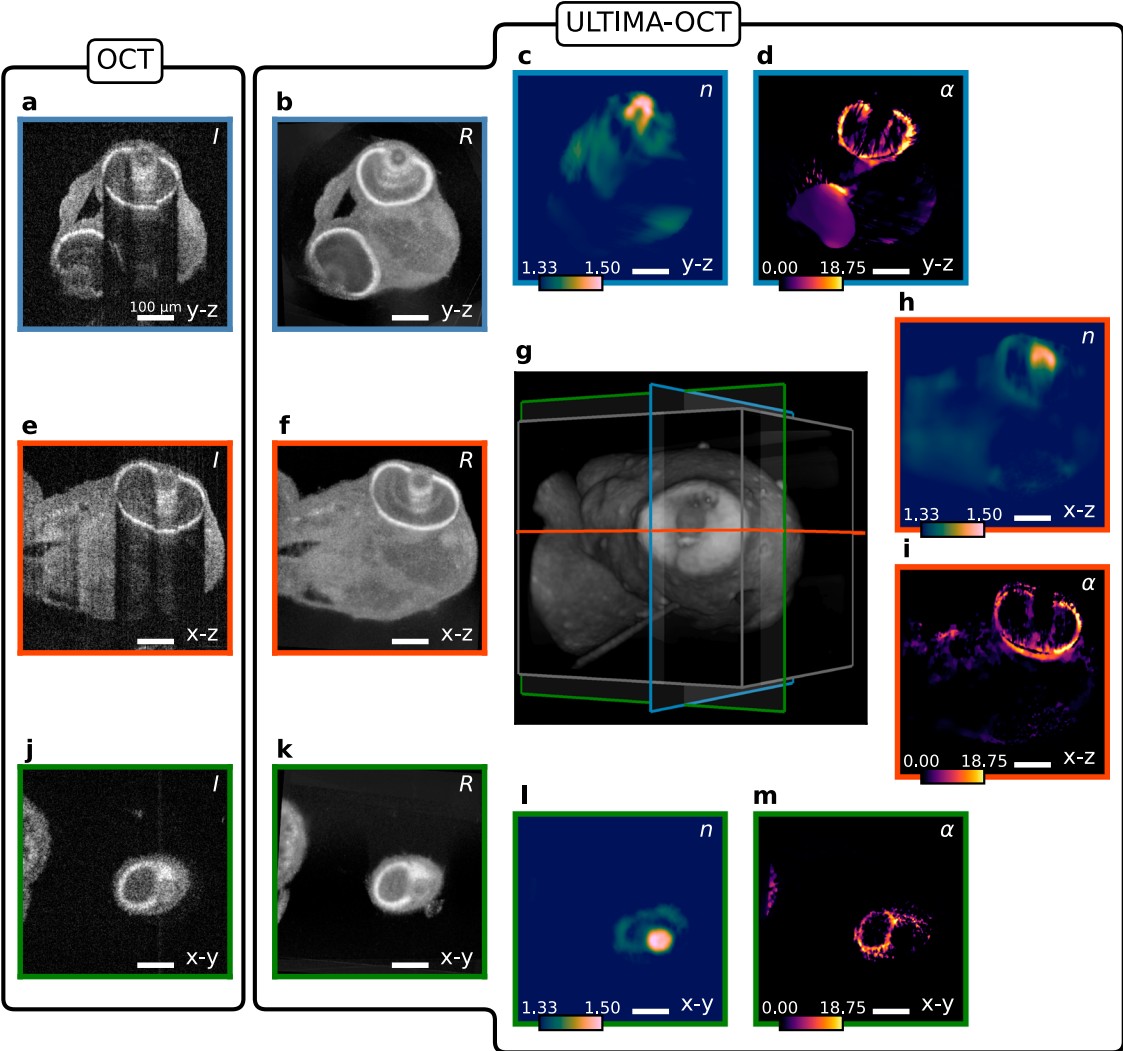

**Fig. 6 | Reconstruction results for the head section of a 3 dpf zebrafish embryo of a mutation with reduced pigmentation. a–d** The (y–z), **e, f, h, i** the (x–z), and **j–m** depict the (x–y) sections of the reconstructions. The leftmost column (**a, e, j**) shows the sections of the recorded OCM volumes in logarithmic scale, whereas the adjacent column (**b, f, k**) shows the reconstructed reflectivity map R of the same sections in logarithmic scale. **d, i, m** Show the slices of the reconstructed attenuation map α (in mm⁻¹), whereas **c, h, l** depict the sections through the reconstructed RI distribution n. In (**g**), a 3D rendering of the reconstructed reflectivity map is shown, together with the planes shown in (**a–f**) and (**h–m**). Scale bars: 100 μm.

stitching. ULTIMA-OCT has an acoustic chamber that can levitate samples up to 1 cm away from the bottom and the top chamber surfaces, preventing these issues. Manipulating samples far from reflecting surfaces might not be possible for alternative acoustic strategies utilizing SAW as such devices are difficult to scale up. Further, we are unaware of stable reorientation of levitated and strongly asymmetric samples such as zebrafish larvae in devices utilizing acoustically induced microstreaming from bubbles, solid structures, or SAW, although demonstrated for the more axis-symmetric *Caenorhabditis elegans*[48].

To avoid motion artifacts in OCT imaging of live zebrafish anesthesia is necessary. We do not anticipate any issues in achieving stable trapping of live sedated fish, since behavioral response under the influence of optical tweezers has been successfully studied under similar conditions[49]. OCT imaging of wake fish embryos does not seem feasible, since the considerably stronger acoustic forces necessary to stall an actively swimming fish will have non-negligible bio-effects (such as heating and cavitation), besides the problem of motion artifacts that will be inevitable. The acoustic radiation forces proportional to the gradient of the acoustic potential[39], lead to a sufficient trap-

stiffness to confine our investigated samples on the order of or smaller than λ/4. Importantly, between rotations and during the acquisition times of 8–23 s to capture a single angle volume of a field of view ranging from 0.67 × 0.67 to 0.76 × 3.78 mm, the sample could be kept stable by acoustic trapping, i.e., we did not observe any motion artifacts.

The shape of the levitated specimen has an influence: for elongated and very asymmetrical samples such as the zebrafish embryos, we found that we can precisely control the orientation to basically any desired angle, as seen in Supplementary Movie 2. The induced rotation angle is proportional to the step size in the adjusted voltage or phase, and by choosing a smaller step size than the 1 V used in this example, we can achieve finer precision. Reorientation leads to rotation around the major axis, but additionally, there can also be a tilt due to asymmetry in the mass distribution and due to non-uniform acoustic forces in the chamber. The change in the acoustic trapping landscape to reorient the sample can also lead to small translations in any direction before the object again is stably trapped in the new local pressure minima. The translations and rotations are in general reversible when reverting the acoustic settings and small orientation changes, lead to

small translations. Along the $y$-axis in the chamber, where we do not propagate acoustic waves, the sample is confined by the transverse component of the acoustic radiation forces from the (non-ideal 'plane') standing waves in the orthogonal directions[50], and during rotations, we generally observe larger drifts than in $x$- and $z$-direction where the trap-stiffness is higher. See Supplementary Methods for more details. The change in position and tilt is dealt with by our reconstruction algorithm and does not impose a serious restriction. For time efficiency in imaging, we adjust the field of view between re-orientations to permit as small imaging volume as possible, and for the few significant translations along the $z$-axis, we adjust the $z$-position of the sample stage.

Since the targeted types of samples are never perfectly spherical, our manipulation strategy is also suitable for less asymmetric samples as shown in Supplementary Fig. 4 and Movie 3, where we successfully manipulated melanoma spheroids. In order to handle organoids in our platform, we would need the organoids in suspension and this will be a focus in future work. It has already been demonstrated that acousto-fluidics can play a useful role in the trapping and merging of organoids[51], where organoids were formed in suspension or in Matrigel followed by Matrigel removal before acoustic manipulation and further growth in gel. Recent efforts have led to the development of alternative scaffold-free growth of several types of organoids in suspension[52,53]. Organoid growth mediated by ultrasound may also be possible, similar to growing cancer spheroids under acoustic trapping[54–56].

The range of trappable sample sizes depends on the design of the acoustic chamber. In previous studies, we found that we can control the transient and sustained rotation of samples up to a thickness around $\lambda/2$[11,16]. Our earlier platforms implementing acoustic trapping around 3 MHz were limited to the manipulation of samples up to roughly 200 μm in size. These devices were restricted to trapping close to the bottom coverslip and the induced transient rotations were typically limited to 90°. The current work provides a solution to these limitations and facilitates the manipulation of mm-sized samples at 0.6 MHz and 2.5 mm wavelength in water. The symmetry of the current chamber allows for excitation of the same orthogonal resonance modes which facilitates 360° step-wise rotation with two transducers. For large samples relative to the trapping wavelength, the acoustic radiation forces scaling with the sample radius becomes more complicated[38,40] than in the small particle limit[57]. However, we are not at the limits in driving voltage and we believe our platform is also suitable for handling larger samples up to a thickness of above 1 mm. Concerning the lower bound for sample sizes, we have demonstrated trapping and reorientation of a cancer spheroid of about 400 μm thickness. The occurrence of sustained rotations depends on the sample shape and could be suppressed for our investigated samples, but the exact limits of size and shape need further investigation. If higher trap-stiffness is found necessary to manipulate significantly smaller sample sizes, one could either explore using a higher harmonic frequency in the same device, or transducers of higher resonance frequency, which we elaborate on in the Supplementary Methods along with details on the choice of transducers and size of the acoustic chamber.

A multitude of similar acoustic trapping platforms[17,58,59] have verified the biocompatibility of acoustic trapping, and our measured pressures are below the limits of which adverse bio-effects and cavitation are expected. However, we will need to assess the effects of our specific acoustic trapping platform for long-term experiments on the targeted live samples in future experiments. The large sample volume in our chamber minimizes the effects of a potential harmful influence of the materials and glues used in this prototype and as bio-compatible glues and coatings are available, this does not pose a future concern. Details on how our acoustic chip can be modified to make it suitable

for long-term monitoring of life samples with biocompatibility testing can be found in the Supplementary Methods.

The targeted samples are addressable by OCM, with the penetration depth depending on the light source's wavelength, the sample's scattering, and attenuation properties. Longer wavelengths allow for deeper penetration at the cost of lower resolution. High-scattering and high-attenuation structures can obstruct the incident beam, resulting in shallower penetration. Therefore, OCT imaging depth is limited to 1–2 mm for most biological samples, and in OCM it can even be less. To achieve high lateral resolution, the NA is increased in OCM. However, this can result in a decrease in the depth of field and lead to non-uniform lateral resolution at varying depths. Ultrasound imaging can reach deeper but has a poorer resolution. ULTIMA-OCT maintains the high resolution of OCT/OCM and compensates for shadow artifacts, thereby reconstructing the sample beyond the limits of a single angle's penetration. In this work, the OCM beam did not fully fill the objective aperture. This resulted in a larger depth of field (as shown in Supplementary Fig. 7), which worked for zebrafish larvae. However, to achieve higher reconstruction accuracy for large samples, interferometric synthetic aperture microscopy[60] or other computational aberration correction[61,62] could be investigated in the future. The axial phase stability of the ULTIMA-OCT was characterized (as shown in Supplementary Fig. 8). By comparing with the data of a zebrafish embedded in Phytagel, it is found that the ULTIMA-OCT axial phase instability originated mainly from the OCM system rather than from acoustic trapping. There was a small axial fluctuation (within 10 nm) of the acoustically trapped zebrafish, but it did not affect the image acquisition and reconstruction. This makes the adaption of the acoustic chamber to functional OCT such as OCT angiography, Doppler OCT, spectroscopic OCT, and optical coherence elastography[63] possible to provide further biological information. Our acoustic actuation strategy might also be combined with other imaging techniques such as fluorescence-, multi-photon-, photoacoustic microscopy, and Raman spectroscopy, also in conjunction with OCT in dual-/multi-modal systems for more applications at different organization levels[63]. Our future work will also include investigations of simplified chamber designs with a square cross-section and two orthogonal transducers which would open up for manipulation closer to the bottom coverslip if desired for the targeted imaging setup.

The tomographic reconstruction of reflectivity, attenuation, and RI, performed in this work and explained in the "Methods" section below, uses fewer viewing angles compared to previous work on samples immobilized in gel[37,45]. However, the greater uncertainty in the orientation of the sample in our case adds additional ambiguity, which makes the reconstruction process even more challenging. To achieve sufficient accuracy on the registered angles we made use of prior information in the reconstruction process. In the first stage of the algorithm, it is crucial to make heavy use of regularization on the reflectivity map $R$ to deal with the unknown reorientation in between volume recordings. To make this step efficient, coarser representations of the recorded data can be used to obtain a first guess of the reflectivity, RI, attenuation map, and motion. After the motion is registered with sufficient accuracy, the high-resolution reconstruction can be initialized with the parameters estimated from the first stage. Additionally, the strength of the regularization (explained in the "Methods" section and in more detail in the Supplementary Methods) can be lifted in order to also record fine-grained structures and to make use of the full resolution in the OCT dataset.

Limitations of the presented reconstruction approach concern the attenuation and RI maps. In modeling the attenuation we assumed the scattering and absorption to be independent of the recording angle. While the attenuation map is a useful quantity for the fidelity of our model, the angular independence might not be given and therefore the reconstruction of $\alpha$ is limited in its informative value. As the

number of orientations used in this work is small, the reliability of the RI map outside the eye regions may be restricted. Although the RI values of the lens portion of the eye can be estimated accurately, the reconstruction of the RI map is strongly dependent on the available angular coverage in the dataset and the specimen itself. As a change in the RI at a location manifests itself as a delay in the structures behind that location, distinct structures have to be visible in the OCT signal. Since the reconstructed maps for reflectivity, attenuation, and RI are tightly linked physically, future work on image fusion could entail utilizing a more refined physical forward model in the form of a unified treatment of those quantities. Our strategy of multi-angle OCT could potentially also profit from a full wave-optical treatment of the light-matter interaction[64].

ULTIMA-OCT combines cutting-edge modalities in acoustic actuation, OCT, and model-based tomographic reconstruction. Our 3D printed low-cost chamber is a simple add-on to an OCT imaging platform permitting multi-angle acquisition of a large range of samples. The presented strategy of step-wise reorientation, registration, and reconstruction can be applied to a wide range of microscopy techniques. OCT is particularly suitable to give insight into dynamics and morphological changes in live biological samples due to its label-free and non-invasive nature. Through genetic overlap with model organisms or human-derived in vitro models, long-term monitoring of these samples provides valuable insights into human disease. The advancement of patient-derived spheroid and organoid models holds promise for personalized medical diagnosis and treatment, and for acceleration of oncology studies, for instance, compared to animal models. Our acoustic chamber not only offers a solution for non-contact manipulation of these samples, but the multi-angle OCT acquisition combined with our reconstruction algorithm enables a detailed reconstruction of samples at an enhanced penetration depth. Consequently, ULTIMA-OCT permits 3D reconstruction of larger or more optically dense samples and we believe this technique holds great potential in biomedical research for quantitative analysis of developing structures to distinguish and quantify for instance volume and surface of internal structures and voids or necrotic regions and for tracing effects from drugs or genetic modifications.

## Methods

### Acoustic manipulation chamber

The chamber frame with a symmetric octagonal cross-section is 3D-printed (Original Prusa i3 MK3, Prusa Research, Czech Republic) in a polymer (PET-G, RS: 891-9309) with open front- and backside and with windows around the eight sides for attaching four piezo-electric transducers and four reflectors. The four transducers are positioned on the top part of the chamber, as shown in Fig. 2b and Supplementary Fig. 1 (top-transducer and S1–S3 side transducers) with the four reflectors on the opposite parallel side. For imaging compatibility through the bottom of the chamber, a 170 μm thick coverslip seals the bottom and acts as the reflector of the acoustic waves from the top transducer. The remaining three reflectors (R1–R3 in Supplementary Fig. 1) are machined in aluminum or cut from a 170 μm coverslip. All four transducers are (8 mm × 15 mm, 3 mm thick) plate transducers made of Lead Zirconate Titanate (PZT) (Pz26, CTS Ferroperm, Denmark). With the aim of levitating samples of a size in the mm range, we chose these transducers with a thickness resonant mode frequency of the bare transducer around 670 kHz (wavelength in water of about 2.2 mm). Resonantly enhanced BAW generates standing waves of sufficient force in our chamber to levitate and reorient our samples. The specific chamber height used here (19.2 mm) was found by an iterative approach of adjusting the chamber dimensions of the 3D printed frame based on simulations and characterizing the acoustic resonances by electrical impedance measurements and experiments. See Supplementary Fig. 2 for specific chamber dimensions and detailed information in Supplementary Methods.

The backside of the printed chamber frame is covered by an octagon-shaped aluminum plate, to seal the back of the chamber. To attach the parts to the printed chamber frame, we use cyanoacrylate glue followed by nail polish to completely seal all remaining gaps. The bottom silver-plated electrode of each piezo-plate is connected to the aluminum with silver paint (RS: 123-9911) for thermal and electrical connection (common ground). To electrically connect to the top and the bottom electrodes on each transducer. We use copper wires and silver paint for the top electrode and aluminum plate, respectively (Supplementary Fig. 1). We drive each transducer with a sinusoidal signal from waveform generators: two single output waveform generators (Agilent 33220A) to drive side transducers S1 and S2 (at 590 kHz), and one dual output waveform generator (Keysight 33522B) to drive the top and S3 transducer (at 600 kHz). Each signal is amplified by power amplifiers (EVAL-ADA4870, Analog Devices) and impedance-matching transformers, see details in Supplementary Methods. To ensure levitation of our samples we operate the top transducer above 20 V, and to reorient the samples we tune the voltages of the transducers in the range of 20–35 V. This corresponds to maximum pressure amplitudes in the range of 80–150 kPa measured in the anti-node with a hydrophone (NH0200, Precision Acoustics, United Kingdom). Please note that voltage refers to peak-to-peak voltage throughout the paper.

### Operation for multi-angle imaging

The assembled octagon chamber is placed in a 3D-printed sample holder (Supplementary Fig. 1) that fits on the inverted microscope stage, attached via an adapter above the objective in the imaging setup in the case of OCT. We tilt the holder 90° so the open chamber front faces upwards and rinse the chamber with a 0.1% Triton X-100 solution (Sigma-Aldrich) to make the chamber more hydrophilic, to limit bubble formation at surfaces when filling the chamber. We then fill the chamber with the liquid [distilled water, tap water, or 1X phosphate-buffered saline (PBS)], place the sample inside, and seal the front with a coverslip. This front coverslip is kept in place by adhesion forces, and can easily be removed for sample exchange. The top transducer is turned on and we tilt the chamber to levitate the sample in the middle region of the chamber. The holder is placed on the imaging stage, and we start the acoustic manipulation to reorient the sample and acquire images through the bottom coverslip of the chamber at each desired step. Darkfield image acquisition (see Supplementary Methods for setup details) was used before performing OCM to optimize the acoustic settings for stable reorientation. The acoustic rotation and OCM imaging procedure is illustrated in Supplementary Movie 1 and Supplementary Fig. 3. The top transducer is "far away" from the trapped sample (about 7–12 mm), but to further limit the back-reflection from the top transducer during OCM, we paint the bottom silver-plated electrode black, which does not affect its acoustic performance.

### Samples

**Zebrafish (*Danio rerio*) embryo preparation.** In this work, we used zebrafish of the pigmented wild-type Tubingen strain (wild-type) and double mutant transparent *Mitfa^{b692/b692}/ednrb1^{b140/b140}* fish with reduced melanophores and iridophores (the second zebrafish line enabled deeper penetration and better anatomy visualization, and was used in this work for comparison with the wild-type). After spawning, eggs were maintained in egg water at 28 °C under standard conditions for up to 3 and 5 dpf. After overnight fixation in 4% paraformaldehyde (PFA), embryos were washed with PBS and stored at 4 °C until they were used for imaging.

**Spheroid preparation.** The murine melanoma cell line B16-F10 was used to form spheroids by the hanging drop method. Cells were grown in Dulbecco's modified Eagle medium supplemented with 10% fetal bovine serum, 1% penicillin/streptomycin in a humidified incubator at

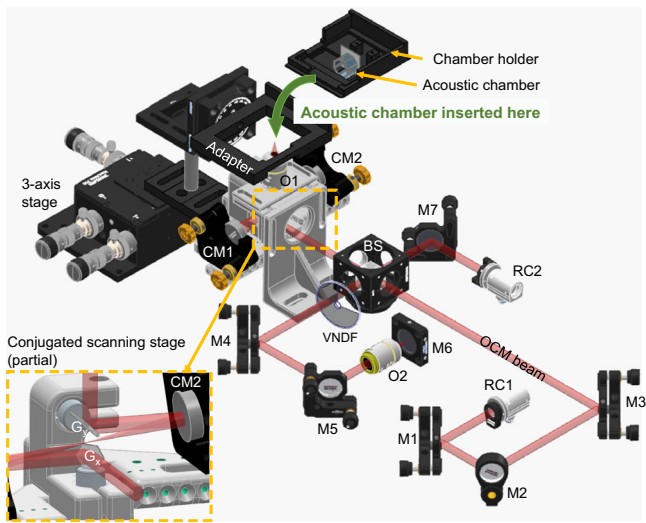

**Fig. 7 | The schematic of the OCM system and the add-on acoustic chamber.** RC reflective collimator, M mirror, BS beam splitter, VNDF variable neutral density filter, CM concave mirror, O objective, $G_x$ $x$ galvanometer scanner, $G_y$ $y$ galvanometer scanner. Credit to Thorlabs Inc. for drawings of optical components (RC04APC-P01, PF10-03-P01, CM254-050-P01, BS065, B4C/M, C4W, KM100, KS05T/M, LMR05/M, TRF90/M, CP35/M, NDC-50C-4, RMS10X, MAX313D/M, RP01/ M, BA2/M, AP90/M, TR75V/M).

37 °C, and 5% $CO_2$. When 80% cell confluency was reached, cells were trypsinized, and 1000 cells per 25 µL media were placed as droplets onto a Petri dish lid, inverted, and returned as top of the Petri dish bottom part, which was filled with 15 mL PBS for humidity and incubated for 4 days. During this time, individual spheroids formed in each droplet. The spheroids were collected in microcentrifuge tubes, washed three times with PBS between centrifugation (300 × $g$, 5 min), and fixed with 4% PFA at room temperature for 20 min before washing three times again with PBS. Spheroids were then stored at 4 °C until use.

### Adaptation of OCT set-up

The setup[65] (as illustrated in Fig. 7) used in this work is a spectral domain OCM system using a compact polarization-aligned three-superluminescent-diode laser source (EBD290002, EXALOS AG)[66]. The OCM laser source had a center wavelength of 845 nm and a wide bandwidth of 131 nm, resulting in a high axial resolution of approximately 3.7 µm in air, corresponding to 2.68 µm in tissue (with an $RI$ of 1.38). The OCM lateral resolution was around 3.4 µm, and the depth of field (defined by twice the Rayleigh length, see Supplementary Fig. 7) was approximately 153 µm in air and 211.1 µm in tissue. Through reflective collimator (RC04APC-P01, Thorlabs) 1, the OCM beam was directed to the system and then divided by a beam splitter (70:30 (R: T), BS065, Thorlabs) to the sample arm and reference arm, respectively. The OCM laser was focused on the sample and the reference mirror using objectives (CFI Plan Fluor 10×, Nikon, NA = 0.3), and the laser power on the sample was around 1.53 mW. Reflective collimator (RC04APC-P01, Thorlabs) 2 was connected with a homemade spectrometer[66,67] to capture the interferogram of backscattered light from the reference mirror and the sample. The acoustic chamber was mounted on a chamber holder and implemented in the OCM system using a 3D-printed adapter. Precise sample positioning and focus adjustment were achieved using a three-axis translation stage (MAX313D/M, Thorlabs).

Volumetric OCM data was obtained by the raster scanning of a pair of galvanometer scanners (CTI6220H, Cambridge Technology) inside a conjugated scanning stage. During imaging, a 20 kHz camera line scan rate of the OCM spectrometer was used, corresponding to a sensitivity of 104.7 dB. For zebrafish embryo imaging, a scanning step size of 1.68 µm was used for smaller field-of-view imaging (head region), and 2.52 µm was used for whole fish imaging. A scanning step size of 1.68 µm was employed for melanoma spheroid imaging.

After standard OCT preprocessing steps (background subtraction, resampling, digital dispersion compensation, fast Fourier transform, and logarithmic calculation), OCM raw binary data was converted to three-dimensional images[68]. En face images of the zebrafish embryos were obtained by average intensity projection, and the en face image of the melanoma spheroid was obtained by standard deviation projection using Fiji[69]. To create the OCM cross-section image of the zebrafish embryo, seven B-scans were averaged consecutively. Similarly, the OCM cross-section image of the melanin spheroid was obtained by averaging three consecutive B-scans. 3D rendering of the volumetric data was achieved using Amira 3D (Thermo Fisher Scientific, version 2023.1.1).

### Reconstruction algorithm

The dataset consisting of OCM volumes of the specimen at multiple different orientations serves as the starting point for the reconstruction algorithm. We follow a model-based approach to describe the observed OCM data by the interaction of a reflectivity $R$, attenuation $\alpha$, and $RI$ contrast map $\Delta n$. Inspired by the works of Zhou et al.[37,45] and Vermeer et al.[70] the detected OCM signal $I$ is modeled line-wise by a layer-by-layer based propagation

$$I_i = R(\mathrm{r}_i) \cdot H(z_i) \cdot T(z_i) \cdot \exp\left(-2\sum_{j=0}^{i} \alpha(\mathrm{r}_j)\Delta z_j\right) \quad (1)$$

$$z_{i+1} = z_i + \frac{\Delta z}{n_0 + \Delta n(\mathrm{r}_i)}. \quad (2)$$

Starting with $\mathrm{r}_0$ as the boundary conditions for the coordinates, the signal is traced through the specimen represented by $R$, $\alpha$, and $\Delta n$. $T$ and $H$ denote the confocal point spread function and sensitivity roll-off, respectively[71].

To extract the maps for $R$, $\alpha$, and $\Delta n$ we formulate the reconstruction as an optimization problem. The error metric we aim to minimize is composed of data fidelity, TV, and $l2$-norm, as well as positivity constraints on $R$, $\alpha$, and $n$. In addition to $R$, $\alpha$, and $n$, also $q$, and $t$, rotation parameterized by quaternions and translations, represent optimization parameters. We solve the optimization problem jointly with Stochastic Gradient Descent (SGD), where we first extract motion parameters and reconstructions on a low-resolution representation. Afterwards, the high-resolution reconstruction is initialized with the resulting obtained low-resolution quantities, and further refined iteratively to yield the final reconstruction. For a detailed explanation, we refer to Supplementary Fig. 6 and Supplementary Methods.

The numerical optimization was conducted in JAX[72] 0.4.13 using Python 3.11 on a workstation equipped with an Nvidia RTX 4090 GPU. To obtain the low-resolution reconstructions we ran the algorithm for 1000 iterations and used a 3× smaller resolution ($134^3$ and $150^3$), whereas for the high-resolution reconstructions ($400^3$ and $450^3$) 250 iterations were performed. An iteration represents a single pass through the entire dataset. The whole reconstruction process took approximately 20 min.

### Reporting summary

Further information on research design is available in the Nature Portfolio Reporting Summary linked to this article.

## Data availability

The datasets supporting the findings in this study are available from the corresponding author upon request. Image and reconstruction data related to displayed items are available at https://github.com/simo343/ultimaoct_data. Source data are provided with this paper.

## Code availability

The source code for the ULTIMA-OCT reconstructions is available at https://github.com/simo343/ultimaoct.

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

## Acknowledgements

This work was funded in part by the Austrian Science Fund (FWF) SFB 10.55776/F68 *Tomography Across the Scales*, projects F6803-N36 Multi-Modal Imaging (WD) and F6806-N36 Inverse Problems in Imaging of Trapped Particles (MRM), and in part by H2020-ICT-2018-20 project REAP with grant agreement ID 101016964 (WD), and the Joint Ph.D. Program Medical University of Vienna/NTU Singapore "Kooperation Singapur" Grant No. SO10300010 (WD). For open access purposes, the authors have applied a CC BY public copyright license to any author-accepted manuscript version arising from this submission. We thank Nicole Schmitner (Institute of Molecular Biology, University of Innsbruck) for providing us with zebrafish embryos, describing the features and sample preparation, as well as for valuable discussions, Abigail J. Deloria (Center for Medical Physics and Biomedical Engineering, Medical University of Vienna), Agnes Csiszar, and Gergely Szakacs from the Center for Cancer Research, Medical University of Vienna for providing us with melanoma spheroids and the sample preparation description. We thank Gregor Thalhammer-Thurner (Institute of Biomedical Physics, Medical University of Innsbruck) for valuable discussions. We would like to express our gratitude to Richard Haindl (Center for Medical Physics and Biomedical Engineering, Medical University of Vienna) for the OCM software development and the homemade spectrometer build. We thank M. Duelk from EXALOS AG for the OCM laser source. We would like to acknowledge the FWF doc.funds Ph.D. program Image-Guided Diagnosis and Therapy (IGDT) at the Medical University of Innsbruck in which Mia Kvåle Løvmo is enrolled.

## Author contributions

W.D., M.R.M., and S.M. conceived the general idea of the work. M.K.L. developed the acoustic manipulation strategy, and designed and fabricated the acoustic chamber. M.K.L. and S.D. carried out the acoustic-OCT experiments at the Medical University of Vienna. S.D., R.L., and W.D. planned the OCM experiments, S.M. and R.L. developed the numerical reconstruction algorithm, and all authors contributed to the structuring and writing of the paper.

## Competing interests

The authors declare no competing interests.
