## [Peer Review File · Nature Communications]

Ultrasound-Induced Reorientation for Multi-Angle Optical Coherence TomographyREVIEWER COMMENTS

Reviewer #1 (Remarks to the Author):

In this paper, the authors present an ultrasound manipulation device for trapping, rotation, imaging, and reconstruction of samples like zebrafish larvae and melanoma spheroids. The methodology is interesting and offers advantages to enhance optical coherence tomography reconstruction. Overall, the concept of ultrasound-induced rotation is relatively new, the experiments are well-designed. To become publishable in the journal, I would suggest the authors consider the following points.

1. Please clearly state the novelty of this technique compared to the previously published method by the same group (e.g., DOI: 10.1039/D0LC01261K, Lab Chip, 2021, 21, 1563-1578).

2. The manuscript lacks a thorough review of the literature. Notably, there are several publications detailing the rotation of cells, *C. elegans*, and zebrafish embryos that are not mentioned. How does this technique compare to existing methods used for trapping and orienting larvae?

Acoustic-actuating rotation:

[1] SonoRotor: An Acoustic Rotational Robotic Platform for Zebrafish Embryos and Larvae

[2] 3D mechanical characterization of single cells and small organisms using acoustic manipulation and force microscopy

[3] Rotational manipulation of single cells and organisms using acoustic waves

[4] 3D manipulation and imaging of plant cells using acoustically activated microbubbles

[5] Embedded Microbubbles for Acoustic Manipulation of Single Cells and Microfluidic Applications

[6] Acoustofluidic rotational manipulation of cells and organisms using oscillating solid structures

[7] Soft-Contact Acoustic Microgripper Based on a Controllable Gas–Liquid Interface for Biomicromanipulations

[8] Surface acoustic waves enable rotational manipulation of *Caenorhabditis elegans*

Magnetic-actuating rotation:

[9] Dynamic and non-contact 3D sample rotation for microscopy

Others:

[10] High-throughput in vivo vertebrate screening

[11] Three-dimensional reconstruction and measurements of zebrafish larvae from high-throughput axial-view in vivo imaging

Amplitude and phase modulations of acoustics:

[12] Rotation of non-spherical micro-particles by amplitude modulation of superimposed orthogonal ultrasonic modes

[13] Rotation of fibers and other non-spherical particles by the acoustic radiation torque

3. "since the stable trapping positions and orientations in the acoustic force fields to some extent also depend on the unknown sample itself." It is still not clear how the acoustic field trap and rotate the sample. Theoretical explanation and numerical simulation of the acoustic field are helpful for readers to understand.

4. "finely tunable stepwise reorientation" what is the rotation resolution that can be achieved? Is the step spacing constant when the sample is rotated to different orientations?

5. A timing diagram of the voltage modulation is necessary to clearly show the step-wise manner.

6. What is the ultrasound excitation voltage and frequency of each rotational experiment?

7. Could a live zebrafish embryo be rotated with the device? The damages to and viability of the larvae were not tested.

8. In zebrafish studies, multi-directional orientation is needed for the purpose of high-resolution imaging of the organs, tissues, and cells. They can discuss if the method is more useful for morphological studies, organ analysis, tissue investigations, or even interrogations at the cellular level (e.g., fluorescent imaging of neurons).

9. Why the object does not move along the y-axis? Is the rotation coaxial with different orientations?

10. "Supplementary Movie 2 showing reorientation by two transducers of a zebrafish embryo." If two transducers are enough to work, why do you use 4 transducers?

11. What is the model-based algorithm developed in the manuscript? A more detailed illustration with schematics or equations is helpful to highlight the novelty.

12. "an AC signal" is it a square signal or a sinusoidal signal?

Reviewer #2 (Remarks to the Author):

In their manuscript, M. Lovmo and colleagues demonstrate a new contact-free method to achieve multi angle Optical Coherence Tomography in order to allow label-free high-resolution 3D imaging of weakly scattering samples. The authors have created a new relatively simple design to create a multi-element acoustic cavity around a thick sample in order to translate and rotate it upon changes in the acoustic parameters of the piezo elements. This enables to achieve multi-angle OCT, which allows a rough quantification of the sample optical parameters such as the local reflectivity, refractive index, and attenuation, and to recover signal from the bottom of the sample.

Overall, I am quite enthusiastic about this work, which represents a highly skilled experimental realization. The paper is clear, and the authors answer most of the questions I had reading the manuscript in the Supplementary document, and the manuscript seems almost ready for publication. However, it is not 100 % clear to me why their nice technology is important and why other researchers should adopt it? It could be helpful that the authors state more clearly what they had in mind before developing such a solution beyond quite standard applications in the field and why they need such a nice but heavy solution? I also have a few genuine questions and remarks about the paper I'll share below.

To summarize, as for me, this manuscript deserves publication in Nature Communications since it shows a new beautiful technical realization of label free imaging, with a clear and solid methodology, and despite the fact that the overall significance to the field and to biological imaging might be questioned.

General questions and remarks

- In general, the introduction is mostly oriented towards organoid imaging, while most of the results show Zebrafish and spheroids. Do the authors have access to organoids? Could they show the real gain of their approach on organoids? I guess this is the aim of their future research, but I feel the manuscript could gain in impact with an effective and useful application?

Naively, I would have guessed that organoids have a circular symmetry so that optical access to the upper half is of minor interest. I am also personally a bit skeptical on the limitation of using a matrix scaffold described in the introduction. I feel (but I agree it can be discussed) that applying a small confinement is important for normal cell function and organoid growth. Focusing the application on zebrafish mostly in order to avoid the attenuation by the eye is a quite poor model (although very nice to demonstrate the technique), since there are not so many anatomically important structures that are missed or that cannot be imaged (eventually by mounting at a different angle). In the worst case, many studies have shown that zebrafish can live and behave almost normally at this stage after enucleation (although I reckon this is a bit brutal!).

- It would be highly beneficial to make this technology more accessible. Could the authors consider sharing their 3D part designs and more instructions to build their acoustic chamber? It seems that they are also a lot of technical tips that the authors use but not describe extensively. Would it possible to describe all the steps towards imaging. What parameters are tuned in order to center the fish to the center of the chamber and not any acoustic node? How is it possible to find the good parameters for a new sample with unknown symmetry? It seems there is a lot of technical skills to have to enable the use their system, and describing them properly would help making this paper reproducible.

- Could the authors quantify the movements of the sample inside the trap: Is the sample axially stable? Could the authors provide a dynamic phase measurement to quantify axial fluctuations at a fixed trap configuration? Because the acoustic wavelength is so large compared to the axial resolution, I don't understand why it is possible to maintain the sample at a fixed axial position? Could they quantify the movements to convince the readers that high resolution imaging is still possible? On a similar note, it seems in Video 2 that while rotating, the sample moves back and forth? Could the authors explain this phenomenon and quantify the induced movement? How do they correct potential XYZ drifts during rotation without biasing the reconstruction model?

- Could the authors evaluate the viability of their samples (spheroids and/or organoids) inside the chamber? Could the mechanical forces applied to the sample can affect differentiation and growth of the sample? Is the mechanical force asymmetric at the scale of the sample, and could it induce differential growth of different part of the sample? I am also a bit concern by the use of cyanoacrylate and nail polish to seal the glass plates for live imaging, which are known to affect the cells. Possibly, because the chamber volume is quite large, it may be neglectable... but I would rather see direct evidence that some samples could stay alive for several hours/days within the chamber.

Questions and minor comments:

- Introduction l 38-39: 'Matrigel is complex and variable which gives rise to a certain irreproducibility': Do the authors have a reference to support this claim? I am also wondering why they are focusing only on Matrigel, while there are many other potential gels that could be used? Do they know and can they discuss why such a focus (apparently in the community) was made on Matrigel?

- The authors describe the advantage of being far from the glass coverslip. However, in the current configuration, they need to cross about 1 cm of solution, which seems quite important. Do the authors need to compensate physically for dispersion? Would it possible to reduce the size of the chamber to have the sample closer to the coverslip (e.g. to use higher NA objectives)

- How do the mechanical parameters, in particular viscosity, of the fluid influence the trapping? Often, Spheroids and organoids can secrete some molecules which can progressively change the local viscosity. Would it affect the trapping conditions and efficacy?

- L 258: Could the authors describe here why a second line of Zebrafish is used, and why this one in particular? Is there a reason why this particular line is used rather than more typical (as far as I know) Nacre or Crystal phenotypes?

- Reconstruction L.326: Could the authors provide references for the TV and Tikhonov regularization. I must admit that I am not an expert of such reconstruction, but I am missing the rationale for using such regularization? Are they completely standard, or could the authors provide reconstructions without such regularization, or with different ones?

- Movie 4: could the authors explain what is the “filament” that seems to be under the fish? Is it intentionally added? Is it an artefact? If yes, is it common to have similar artefacts?
- Discussion L429 and also related to the first general question: It is not clear to me why the organoids have to be in suspension or in a gel droplet? Why is it complicated to have organoids in suspension? What does bring the gel droplet? Isn't it contradictory with all the efforts the authors made to design an acoustic trap to avoid adding gel around the organoid? Then, I am wondering if organoids can be a realistic application for this system? Could the authors add more comments on these points in the manuscript?
- Discussion L.512: In their work, the authors seem to use a 10X 0.3 NA objective with a depth of field around 10 μm , while they are imaging samples of several hundreds of microns. I understand that the multi-illumination helps, but how is it still possible to image at 100 μm depth at such resolution? Do they experience non-uniform lateral resolution in depth? Do they need to implement solutions such as ISAM and/or computed AO algorithms to compensate the resolution loss?
- - Supplementary figure 5: Scale bars are missing
- Could the authors add the reference of the microscope objective used? I had to guess it from the animation in video 1 to estimate the depth of field..

Response to the reviewers

We thank the reviewers for the great effort they both have put into reviewing our manuscript. Their insightful critical assessment was highly valuable to us to improve our work. We have taken up all suggestions, and also included additional experimental data in Supplementary Information.

In the following we address the Reviewers' concerns point by point. The suggested modifications led to changes throughout the manuscript and the SI. In the attached file below we show the difference between the current and previously submitted manuscript in blue.

Reviewer #1 (Remarks to Authors):

In this paper, the authors present an ultrasound manipulation device for trapping, rotation, imaging, and reconstruction of samples like zebrafish larvae and melanoma spheroids. The methodology is interesting and offers advantages to enhance optical coherence tomography reconstruction. Overall, the concept of ultrasound-induced rotation is relatively new, the experiments are well-designed. To become publishable in the journal, I would suggest the authors consider the following points.

Reviewer Point P 1.1 — Please clearly state the novelty of this technique compared to the previously published method by the same group (e.g., DOI: 10.1039/D0LC01261K, Lab Chip, 2021, 21, 1563-1578).

Reply: In this work we developed a strategy for non-contact handling and label-free imaging of mm-sized bio-medically relevant samples with the goal of providing a non-invasive solution for long-term monitoring and detailed reconstruction. Model-organisms and *in vitro* models such as spheroids and organoids have great potential in biomedical research and were the target samples in this work. Our acoustic trapping platform was designed to accommodate mm-sized samples as these models become very interesting as they develop and reach this size. While imaging of smaller samples with ODT has been a focus previously, OCT was chosen in this work as it provides a suitable penetration depth at a good resolution. To make our acoustic strategy compatible with scanning-based OCT, we developed our strategy for acoustic step-wise rotation and designed our chamber to levitate samples far from the bottom coverslip. The development of this strategy was accomplished by leveraging on previously gained knowledge in our earlier work. The symmetry of the current chamber allows us to excite the same orthogonal resonance modes, which enables the increased control of sample re-orientations in the full range of 360° compared to previous chambers (limited to 90° rotations typically). The developed reconstruction algorithm capable of handling the uncertainties in the imaging angle is a major advancement in this work. We hope to have answered this request sufficiently by modifying a paragraph in the Discussion which now reads:

The range of trappable sample sizes depends on the design of the acoustic chamber. In previous studies, we found that we can control transient and sustained rotation of samples up to a thickness around $\lambda/2$ (Løvmo et al., 2021, 2022). Our earlier platforms implementing acoustic trapping around 3 MHz were limited to manipulation of samples up to roughly 200 μm in size. These devices were restricted to trapping close to the bottom coverslip and the induced transient rotations were typically limited to 90° . The current work provides a

solution to these limitations and facilitates manipulation of mm-sized samples at 0.6 MHz and 2.5 mm wavelength in water. The symmetry of the current chamber allows for excitation of the same orthogonal resonance modes which facilitates 360° step-wise rotation with two transducers.

(...) The occurrence of sustained rotations depends on the sample shape and could be suppressed for our investigated samples, but the exact limits of size and shape need further investigation.

Further, we do refer to our previous work in several instances also in the Supplementary Methods.

Reviewer Point P 1.2 — The manuscript lacks a thorough review of the literature. Notably, there are several publications detailing the rotation of cells, *C. elegans*, and zebrafish embryos that are not mentioned. How does this technique compare to existing methods used for trapping and orienting larvae?

Acoustic-actuating rotation:

- [1] SonoRotor: An Acoustic Rotational Robotic Platform for Zebrafish Embryos and Larvae
- [2] 3D mechanical characterization of single cells and small organisms using acoustic manipulation and force microscopy
- [3] Rotational manipulation of single cells and organisms using acoustic waves
- [4] 3D manipulation and imaging of plant cells using acoustically activated microbubbles
- [5] Embedded Microbubbles for Acoustic Manipulation of Single Cells and Microfluidic Applications
- [6] Acoustofluidic rotational manipulation of cells and organisms using oscillating solid structures
- [7] Soft-Contact Acoustic Microgripper Based on a Controllable Gas-Liquid Interface for Biomicromanipulations
- [8] Surface acoustic waves enable rotational manipulation of *Caenorhabditis elegans*

Magnetic-actuating rotation:

- [9] Dynamic and non-contact 3D sample rotation for microscopy

Others:

- [10] High-throughput in vivo vertebrate screening
- [11] Three-dimensional reconstruction and measurements of zebrafish larvae from high-throughput axial-view in vivo imaging

Amplitude and phase modulations of acoustics:

[12] Rotation of non-spherical micro-particles by amplitude modulation of superimposed orthogonal ultrasonic modes

[13] Rotation of fibers and other non-spherical particles by the acoustic radiation torque

Reply: Thank you for pointing out these publications to us. We were familiar with most of them - and some were actually already cited in the original manuscript. However, the number of citations is limited, and as we wanted to have a balance between the fields of OCT, acoustics and inverse problems in the Introduction, we concentrated on the actual (non-contact) scenario we use, i.e. BAW (bulk acoustic waves) and not SAW (surface acoustic waves) or bubble-mediated acoustic generation.

Although many acoustic trapping solutions exist, most are inducing sustained rotations and offer manipulations close to the chamber boundaries which would not be optimal for OCT scanning. Our choice of BAW eases device up-scaling and our re-orientation strategy opens up for OCT imaging. We have now extended the information on alternative methods used in previous works in the main text and in the SI.

We now write in the Introduction:

To tackle larger biological samples, ultrasound techniques for levitation and actuated handling have been developed: Standing bulk acoustic waves (BAW) operating at (sub-)MHz frequencies push biological cells into low-pressure regions (planes, lines or spots depending on the number of orthogonal standing waves) (Thalhammer et al., 2011; Løvmo et al., 2021; Yang et al., 2019), and by modulation of the acoustic waves, objects can be transiently or continuously rotated (Schwarz et al., 2015; Lamprecht et al., 2015; Marzo et al., 2015; Baresch et al., 2018; Løvmo et al., 2022). Surface acoustic waves (SAW) can also be used, e.g., to create acoustofluidic rotational tweezers for morphological phenotyping of zebrafish larvae (Chen et al., 2021). Streaming vortices generated by acoustically induced oscillations of bubbles or solid structures have been applied to rotate zebrafish embryos (Zhang et al., 2023), single cells, pollen-grains and nematodes (Ahmed et al., 2016; Läubli et al., 2021; Ozcelik et al., 2016; Laubli et al., 2021). For compatibility with scanning-based imaging modalities, however, BAW operation is favorable, as it supports tilting of the sample into various stationary (non-rotational) orientations. Bulk waves also simplify device up-scaling for manipulation of large samples far from the chamber boundaries. Scaffold-free confinement in the center of the chamber represents a big advantage of acoustic levitation, since it makes the system more open to perform various assays, such as irradiating parts of the sample by light, adding chemicals and pharmaceuticals by micro-fluidics, or mechanical probing with tips.

Further, in the SI we now mention a few contact-based alternatives to acoustic trapping on p.19 which reads:

(...) Future simulations and experiments could guide a more automated process for the rotational manipulation. One could potentially implement sample orientation detection from images, similar to what is done in the automated, but contact-based rotation for zebrafish

larvae confined in capillaries in VAST (Guo et al., 2017). Other alternatives to acoustically induced rotations usually also involve contact-based sample confinement in capillaries filled with gel (Heintzmann and Cremer, 2002; Van Rooij and Kalkman, 2019) and/or magnetic particles (Berndt et al., 2018), which may alter biological samples morphology.

Reviewer Point P 1.3 — “since the stable trapping positions and orientations in the acoustic force fields to some extent also depend on the unknown sample itself. ” It is still not clear how the acoustic field trap and rotate the sample. Theoretical explanation and numerical simulation of the acoustic field are helpful for readers to understand.

Reply: In order to make it more clear, we have added a reference to previous work to this sentence, where we go into detail of the effects of the sample shape on the acoustic radiation force:

(...) since the stable trapping positions and orientations in the acoustic force fields to some extent also depend on the unknown sample itself (Løvmo et al., 2021).

In addition to the acoustic actuation description in the result section about tuning the magnitude of the restoring torque (also called acoustic radiation torque in the literature) in each direction to reorient an asymmetric sample, we also go into more details in the Supplementary Methods. From p. 13-18, we cover theory of the acoustic torques and both the 4- and 2- transducer approach.

Reviewer Point P 1.4 — “finely tunable stepwise reorientation” what is the rotation resolution that can be achieved? Is the step spacing constant when the sample is rotated to different orientations?

Reply: The rotation resolution depends on the step-size in the voltage or phase adjustment. We have now included this information in the Discussion, which now reads:

The induced rotation angle is proportional to the step-size in the adjusted voltage or phase, and by choosing a smaller step-size than the 1 V used in this example, we can achieve a finer precision.

In the SI on p. 15, where we further discuss the SI Movie 2, we have now added more information in response to the Reviewers questions. It now reads:

(...) Starting from equal amplitudes on the two transducers, we now reduce the strength of the S3 transducer in a step-wise manner and the sample reorients to a more and more dominating top transducer. In this example the voltage is lowered at a step size of 1 V from 33 V down to 19 V and thus one can observe 14 different orientations. By lowering this step-size to 0.1 V, we reorient the sample to a ten-fold more orientations. The orientations will not be perfectly evenly spaced as the acoustic radiation forces acting on the asymmetric sample and the gravitational potential energy of it are dependent on the objects orientation.

Reviewer Point P 1.5 — A timing diagram of the voltage modulation is necessary to clearly show the step-wise manner.

Reply: This is a great idea that we now have included in the SI movie 2.

Reviewer Point P 1.6 — What is the ultrasound excitation voltage and frequency of each rotational experiment?

Reply: The specific frequencies used are found by electrical impedance measurements and experimental verification and remain the same for all experiments. This procedure and numbers are detailed in the SI on p.10 and 13, respectively, and now we have added the driving frequencies of the transducers in the Methods Section:

We drive each transducer with an AC sinusoidal signal from waveform generators: two single output waveform generators (Agilent 33220A) to drive side-transducers S1 and S2 (at 590 kHz), and one dual output waveform generator (Keysight 33522B) to drive the top- and S3-transducer (at 600 kHz). Each signal is amplified by power amplifiers (EVAL-ADA4870, Analog Devices) and impedance matching transformers, see details in Supplementary Methods. To ensure levitation of our samples we operate the top-transducer above 20 V, and to reorient the samples we tune the voltages of the transducers in the range of 20 V to 35 V.

The voltage range of 20 V to 35 V was used in the experiments included in this manuscript, but as we mention in the Discussion, the voltage range used can likely be lowered by a fine-tuning of the settings. This is based on our preliminary data, and will have to be explored in more detail. In the SI we also write about what voltage we typically operated at for initial levitation. We also added more details on specific parameters used to our existing descriptions of the 4- transducer approach in the SI on p.14, now reading:

(...) For the 3 dpf zebrafish larvae a high voltage of 35 V, and a low voltage of 15 V was used.

Further, in the SI, we have now added an example of how to find the parameters for a new sample. In summary, there is not one single sequence that leads to the desired rotation, but observing the sample while adjusting the settings is key. Your great suggestion in the P 1.5 above, now gives more details as well and is a good example of one possible manipulation sequence.

Reviewer Point P 1.7 — Could a live zebrafish embryo be rotated with the device? The damages to and viability of the larvae were not tested.

Reply: From our measurements of the acoustic pressures in our device in comparison to similar studies confirming bio-compatibility, we do not expect any adverse effects due to the used pressures. Due to the biocompatibility demonstrated by others after rapid rotation of zebrafish and high-pressure manipulations of living cells (cited in the manuscript), we believe there will be no problems, but of course biocompatibility is important for our proposed strategy and needs to be assessed. To perform viability studies, we have to first modify our acoustic chamber to use different types of glues and coatings (which was a question from Reviewer#2) and also implement a stable temperature control. In this work, we developed the technique and the algorithm for reconstruction, and the straight-forward temperature

control and glue modifications as well as the more time-consuming, and bio-compatibility assays is a planned next step. We do also mention some strategies such as manipulation at reduced pressures, and at higher frequency if there would be an issue. Regarding trapping capabilities of live actively swimming samples such as zebrafish, we expect trapping of samples under anaesthesia to be no problem. This is performed frequently in imaging and other acoustic trapping strategies of zebrafish. Trapping of swimming samples without sedation will likely not be possible due to the strong forces needed likely leading to unwanted bio-effects. If confinement would be possible, there would likely be motion artifacts during imaging. We have modified a part in the Discussion to make it more clear that trapping of sedated (but not wake) zebrafish is expected to be feasible, and that biocompatibility testing is a next step. It now reads:

To avoid motion artifacts in OCT imaging of live zebrafish anaesthesia is necessary. We do not anticipate any issues in achieving stable trapping of live sedated fish, since behavioural response under the influence of optical tweezers has been successfully studied under similar conditions (Favre-Bulle et al., 2017). OCT imaging of wake fish embryos does not seem feasible, since the considerably stronger acoustic forces necessary to stall an actively swimming fish will have non-negligible bio-effects (such as heating and cavitation), besides the problem of motion artifacts that will be inevitable.

(...)

A multitude of similar acoustic trapping platforms (Bazou et al., 2011; Wiklund, 2012; Chen et al., 2021) have verified the biocompatibility of acoustic trapping, and our measured pressures are below the limits of which adverse bio-effects and cavitation are expected. However, we will need to assess the effects of our specific acoustic trapping platform for long-term experiments on the targeted live samples in future experiments. The large sample volume in our chamber minimizes the effects of a potential harmful influence of the materials and glues used in this prototype and as bio-compatible glues and coatings are available, this does not pose a future concern. Details on how our acoustic chip can be modified to make it suitable for long-term monitoring of life sample with biocompatibility testing can be found in the Supplementary Methods.

We also elaborate on the topic of the measured pressures and expected biocompatibility in the SI on p.11-12.

Reviewer Point P 1.8 — In zebrafish studies, multi-directional orientation is needed for the purpose of high-resolution imaging of the organs, tissues, and cells. They can discuss if the method is more useful for morphological studies, organ analysis, tissue investigations, or even interrogations at the cellular level (e.g., fluorescent imaging of neurons).

Reply: Thanks for bringing up this interesting point. The actuation may indeed be suitable for these types of studies. We might want to investigate these directions in the future.

After the acoustic chamber modification for *in vivo* imaging, ULTIMA-OCT will provide high-resolution, shadow-free 3D monitoring of whole-body morphology development of the zebrafish embryos. The zebrafish brain, eyes, and yolk reconstructed by ULTIMA-OCT contain more details than the standard SD-OCM, which is valuable for studies of these organs. This technique can also be useful for morphological studies combined with organ regeneration, gene modification, disease models, and related drug screening. The OCM resolution in this work was 3.4 μm in lateral directions and 2.68 μm in the axial

direction (in tissue), and they can be further improved by using a higher NA and a broader bandwidth laser source. The resolutions were at the cellular scale, but due to the nature of OCT (based on optical scattering), the cell boundaries cannot be identified as in fluorescence microscopy.

Additionally, OCT is a technique that has many functional extensions (Drexler and Fujimoto, 2008). OCT angiography can visualize vasculature and detect blood flow based on motion contrast. Doppler OCT can measure particle flowing speed based on frequency change. By employing an ultra-broadband laser source or combining several laser sources with different wavelengths, spectroscopic OCT can extract specific molecules (e.g. water, oxyhemoglobin, and deoxyhemoglobin). The acoustic chamber used in ULTIMA-OCT can also be adapted to these functional OCT systems for different organization-level studies.

Furthermore, OCT is an imaging modality that can be combined with other modalities to provide complementary information (Drexler and Fujimoto, 2008). The OCM system used in this work is part of a dual-modality system including OCM and photoacoustic microscope (PAM). Because PAM is based on light absorption, deeper chromophores are usually obstructed by surface absorbers. With the help of the contact-free reorientation of the acoustic chamber, it will be possible to visualize the 3D contribution of the chromophores and improve the axial resolution of PAM limited by the ultrasound transducer (Wang et al., 2019). ULTIMA-OCT can also be combined with multi-photon microscopy, Raman spectroscopy, and fluorescence microscopy for more applications at different scales. We have now summarized this in the Discussion reading:

(...) The axial phase stability of the ULTIMA-OCT was characterized (as shown in Supplementary Fig. 8). By comparing with the data of a zebrafish embedded in Phytigel, it is found that the ULTIMA-OCT axial phase instability originated mainly from the OCM system rather than from acoustic trapping. There was a small axial fluctuation (within 10 nm) of the acoustically trapped zebrafish, but it did not affect the image acquisition and reconstruction. This makes the adaption of the acoustic chamber to functional OCT such as OCT angiography, Doppler OCT, spectroscopic OCT, and optical coherence elastography (Drexler and Fujimoto, 2008) possible to provide further biological information. Our acoustic rotation strategy might also be combined with other imaging techniques such as fluorescence-, multi-photon-, photoacoustic microscopy, and Raman spectroscopy, also in conjunction with OCT in dual-/multi-modal systems for more applications at different organization-levels (Drexler and Fujimoto, 2008). (...)

Reviewer Point P 1.9 — Why the object does not move along the y-axis? Is the rotation coaxial with different orientations?

Reply: Although we do not generate acoustic standing waves in the y-direction in this platform, the samples are still confined in the y-direction as well because of the transverse components of the acoustic radiation forces (ARF) in the orthogonal directions. In our device, the acoustic standing waves are not perfectly plane waves but have pressure variations and local minima where the sample will be stably trapped. Hence, the object is stably trapped within the trapping plane even when only using the top-transducer. The samples do however translate slightly (in all directions, but most notably in y-direction) between re-orientations due to the changes in the pressure landscape, as can be observed in SI movie 2. We have now modified the Discussion and added this information along with a new reference:

(...) Reorientation leads to rotation around the major axis, but additionally there can also be a tilt due to asymmetry in the mass distribution and due to non-uniform acoustic forces in the chamber. The change in the acoustic trapping landscape to reorient the sample can also lead to small translations in any direction before the object again is stably trapped in the new local pressure minima. The translations and rotations are in general reversible when reverting the acoustic settings and small orientation changes, lead to small translations. Along the y -axis in the chamber, where we do not propagate acoustic waves, the sample is confined by the transverse component of the acoustic radiation forces from the (non-ideal 'plane') standing waves in the orthogonal directions (Woodside et al., 1997), and during rotations we generally observe larger drifts than in x - and z -direction where the trap-stiffness is higher. See Supplementary methods for more details. The change in position and tilt is dealt with by our reconstruction algorithm and does not impose a serious restriction. For time-efficiency in imaging, we adjust the field of view between re-orientations to permit as small imaging volume as possible, and for the few significant translations along z -axis, we adjust the z -position of the sample stage.

In the SI on p. 15-16 we now added more details on the drifts between re-orientations and that one could add a third transducer propagating along y -axis if desired.

Reviewer Point P 1.10 — “Supplementary Movie 2 showing reorientation by two transducers of a zebrafish embryo.” If two transducers are enough to work, why do you use 4 transducers?

Reply: This is one of the rare cases, where something worked better than anticipated: We designed this chamber with four transducers to have the possibility to rotate the samples with 4 transducers, because we were not sure whether the two-transducer approach would lead to sustained rotations at the large range of settings as we have observed in previous platforms. However, in this symmetric platform (of much larger size and lower acoustic frequencies) we found that sustained rotations are less likely to occur, and the two transducer approach was sufficient and more time-efficient. We mention our interpretation of these results as follows: In our future work, we will investigate into chambers with a square cross-section and only two transducers as we now have added to the outlooks in the Discussion which now reads:

Our future work will include investigations of simplified chamber designs with a square cross-section and two orthogonal transducers which would open up for manipulation closer to the bottom coverslip if desired for the targeted imaging setup.

We elaborate on the benefits and considerations of future square-cross-section devices with two transducers in the SI on p. 17. Here we have now added the benefit of the 4-transducer approach. This benefit is what led us to the octagon design, but as explained, it was a smaller problem than anticipated. This new part reads:

(...) If sustained rotations turns out to be an issue for targeted mildly asymmetric samples, the 4-transducer approach could be more suited, especially by inducing asymmetries in the chamber to achieve a different resonance frequency for each transducer.

Reviewer Point P 1.11 — What is the model-based algorithm developed in the manuscript? A more detailed illustration with schematics or equations is helpful to highlight the novelty.

Reply: The algorithm we developed is based on a physical model, which numerically simulates the OCM image formation. The model takes as input the reflectivity, attenuation and refractive index as representations of the object. Additional inputs are the different orientations and various imaging parameters (e.g. focal position of the OCM system, alongside other parameters). This model serves as the forward model in an optimization problem, which we solve by a gradient-based approach. To make the developed algorithm more clear, we added a schematic to the supplementary figures (Supplementary Fig. 6), detailing the method.

Reviewer Point P 1.12 — “an AC signal” is it a square signal or a sinusoidal signal?

Reply: We have now edited this sentence to:

(...) We apply a sinusoidal signal to the transducers (...)

And again in the Methods:

We drive each transducer with a sinusoidal signal from waveform generators ...

Reviewer #2 (Remarks to Authors):

In their manuscript, M. Lovmo and colleagues demonstrate a new contact-free method to achieve multi angle Optical Coherence Tomography in order to allow label-free high-resolution 3D imaging of weakly scattering samples. The authors have created a new relatively simple design to create a multi-element acoustic cavity around a thick sample in order to translate and rotate it upon changes in the acoustic parameters of the piezo elements. This enables to achieve multi-angle OCT, which allows a rough quantification of the sample optical parameters such as the local reflectivity, refractive index, and attenuation, and to recover signal from the bottom of the sample. Overall, I am quite enthusiastic about this work, which represents a highly skilled experimental realization. The paper is clear, and the authors answer most of the questions I had reading the manuscript in the Supplementary document, and the manuscript seems almost ready for publication. However, it is not 100 % clear to me why their nice technology is important and why other researchers should adopt it? It could be helpful that the authors state more clearly what they had in mind before developing such a solution beyond quite standard applications in the field and why they need such a nice but heavy solution? I also have a few genuine questions and remarks about the paper I'll share below.

To summarize, as for me, this manuscript deserves publication in Nature Communications since it shows a new beautiful technical realization of label free imaging, with a clear and solid methodology, and despite the fact that the overall significance to the field and to biological imaging might be questioned.

Reply: We thank the reviewer for their kind words in the general assessment. Concerning the question on the relevance of the approach and what our motivation was to undertake this effort, we can say

the following: In fact, the idea for this project was born out of necessity, since OCT imaging of some organoids turned out to be very problematic in another project which involved some of us. We are eager to widen the collaboration and revisit the organoid OCT imaging project, but first we had to develop and demonstrate the method. Which is what we hope to do with this work.

Reviewer Point P 2.1 — In general, the introduction is mostly oriented towards organoid imaging, while most of the results show Zebrafish and spheroids. Do the authors have access to organoids? Could they show the real gain of their approach on organoids? I guess this is the aim of their future research, but I feel the manuscript could gain in impact with an effective and useful application? Naively, I would have guessed that organoids have a circular symmetry so that optical access to the upper half is of minor interest. I am also personally a bit skeptical on the limitation of using a matrix scaffold described in the introduction. I feel (but I agree it can be discussed) that applying a small confinement is important for normal cell function and organoid growth. Focusing the application on zebrafish mostly in order to avoid the attenuation by the eye is a quite poor model (although very nice to demonstrate the technique), since there are not so many anatomically important structures that are missed or that cannot be imaged (eventually by mounting at a different angle). In the worst case, many studies have shown that zebrafish can live and behave almost normally at this stage after enucleation (although I reckon this is a bit brutal!).

Reply: We agree that it would have been nice to also include some results on organoids. We are planning to collaborate with researchers to do so, as the next logical step. However, at this stage we only had easy access to tumor spheroids, but not to organoids grown without any gel or with the gel removed, as we respond to further in P 2.11 below. However, as the acoustic manipulation nicely worked on round cancer spheroids, we expect it to work also organoids not too different in shape and size. It will be interesting to investigate in a next step, where the limitations in size and shape are. You are right that zebrafish can be positioned at various angles, but still this manual positioning is cumbersome and it is difficult to not harm the sample (not to mention enucleation). Furthermore, in order to image Zebrafish larvae with OCT as it matures, grows and become more optically dense, it is necessary to increase the penetration depth to reconstruct the sample. We think this approach, and acoustic trapping in general will be beneficial for both model organisms and *in vitro* models, both regarding its non-contact nature, increased penetration depth and the potential for time-efficient handling. However, we believe non-contact trapping and potentially future growth solves a larger problem for spheroids and organoids which is what we try to convey in the introduction. To make it more clear that the approach is useful for both model-organisms and organoids, we have made minor changes in the introduction, and in the Discussion, we have added examples of useful applications of our approach:

OCT is particularly suitable to give insight into dynamics and morphological changes in live biological samples due to its label-free and non-invasive nature. Through genetic overlap with model organisms or human-derived in vitro models, long-term monitoring of these samples provide valuable insights into human disease. The advancement of patient-derived spheroid and organoid models holds promise for personalized medical diagnosis and treatment, and for acceleration of oncology studies, for instance, compared to animal models. Our acoustic chamber not only offers a solution for non-contact manipulation of these samples, but the multi-angle OCT acquisition combined with our reconstruction algorithm enables a detailed reconstruction of samples at an enhanced penetration depth. Consequently, ULTIMA-OCT permits 3D reconstruction of larger or more optically dense samples

and we believe this technique holds great potential in biomedical research for quantitative analysis of developing structures to distinguish and quantify for instance volume and surface of internal structures and voids or necrotic regions and for tracing effects from drugs or genetic modifications.

Reviewer Point P 2.2 — It would be highly beneficial to make this technology more accessible. Could the authors consider sharing their 3D part designs and more instructions to build their acoustic chamber? It seems that they are also a lot of technical tips that the authors use but not describe extensively. Would it possible to describe all the steps towards imaging. What parameters are tuned in order to center the fish to the center of the chamber and not any acoustic node? How is it possible to find the good parameters for a new sample with unknown symmetry? It seems there is a lot of technical skills to have to enable the use their system, and describing them properly would help making this paper reproducible.

Reply: It is important to us to share our technology, and we have detailed our procedure on several pages in the Supplementary information, which contains chamber dimensions, materials and assembly procedure as well as details on the operation. We could provide the 3D CAD file of the octagon frame on demand. The current sample loading and centering of the sample, is a simple approach that we will modify to make more efficient in the next step. We have added more details about the current and planned strategies for sample loading and centering in the SI on p.13 which now reads:

To manipulate the sample with all transducers, we levitate the sample in one of the five nodes in the vertical direction in the center region of the chamber, where the standing waves from all the transducers intersect. To do this we tilt the chamber filled with liquid and a sample with only the top transducer active. The z-axis of the chamber is horizontal after filling and sealing the chamber, and by tilting the chamber to a vertical position at the high top transducer voltage, the sample usually levitates in the center region. If the sample levitates above the center region, it can be lowered by a rapid decrease and increase in the top transducer voltage, and if it levitates too low, the chamber is again tilted between horizontal and vertical position until the sample levitates in the center region. Usually the sample is levitated in the center region in 1-3 attempts. This method is sufficient for the current proof-of-principle, and in future devices, we plan to implement micro-fluidic inlets and outlets to make loading and unloading of a new sample more efficient. If needed, the inlet to the chamber could consist of a channel with counter-propagating transducers to transport the sample higher or lower by phase-modulation in a 'conveyor belt' fashion (Cox et al., 2022) before it reaches the main chamber. If our future work regarding chambers of a square cross-section with two transducer is successful, the waves from the two transducers could intersect across the whole chamber volume.

Regarding the last question about how we find good parameters for handling a new sample: The key is to observe the sample while changing the parameters in a step-wise manner. Finding the voltage and phase-settings is actually quite straight-forward, compared to the chamber dimension and resonance frequency. We now hope to have clarified this by adding a paragraph in the SI on p. 18-19:

While model organisms at the same development stage are very similar in shape and size, spheroids and organoids typically vary more within the same batch. However, a similar approach can be used across a large range of samples and by direct observation of the sample, the settings can be adjusted on the fly to yield the desired orientation. (...) For a new sample, we determine the minimum levitation voltage by lowering the voltage until the sample sinks. We have not yet discovered the limits of sample weight or size that we can levitate as the 3-5 dpf zebrafish larvae and melanoma spheroids samples we have investigated so far all levitate above 20 V on the top transducer. In this work we typically set the top transducer voltage to 10 V above this limit for rapid initial levitation after sample loading. By observing the sample while increasing the voltage of the orthogonal side transducer (at 0° relative phase) we then find the settings at which the sample reorients its major axis to the y-axis. To keep the rotation around this axis, this is the minimum driving voltage of the side-transducer. From this point, several combinations of phase- and voltage adjustments result in the desired rotation. For instance, we can from here on, lower the top transducer voltage or increase the side-transducer voltage further at a desired step-size to the point at which the forces are similar in the two directions leading to a 45° rotation of the zebrafish larvae from initial (top transducer) orientation. In this symmetric chamber this is close to the same voltage settings of the two transducers. Now, we can lower the top transducer voltage until the sample's minor axis is oriented to the side-transducer. From this position, the phase can be adjusted in the direction that yields the desired rotation (by observation), and again the top-transducer voltage can be increased. Sustained rotations are more likely to occur the more symmetric the samples are and for similar forces from the two transducers. Note that the rotation mechanism does not work for perfectly symmetric samples as the restoring torque vanishes. At equal voltages, the spinning torque can be reduced by adjusting the phase-shift, or one can keep the restoring torque in one direction dominant by avoiding operating at equal voltages and still rotate the sample 360° around its major axis. Future simulations and experiments could guide a more automated process for the rotational manipulation. One could potentially implement sample orientation detection from images, similar to what is done in the automated, but contact-based rotation for zebrafish larvae confined in capillaries in VAST (Guo et al., 2017).

Reviewer Point P 2.3 — Could the authors quantify the movements of the sample inside the trap: Is the sample axially stable? Could the authors provide a dynamic phase measurement to quantify axial fluctuations at a fixed trap configuration? Because the acoustic wavelength is so large compared to the axial resolution, I don't understand why it is possible to maintain the sample at a fixed axial position? Could they quantify the movements to convince the readers that high resolution imaging is still possible? On a similar note, it seems in Video 2 that while rotating, the sample moves back and forth? Could the authors explain this phenomenon and quantify the induced movement? How do they correct potential XYZ drifts during rotation without biasing the reconstruction model?

Reply: Acoustic gradient forces determined by pressure gradients drive the sample into a stable trapping position around a local pressure minimum, and samples much smaller than the wavelength are typically manipulated in acoustic trapping platforms. We do not observe any motion artifacts in the OCT imaging, or in our reconstructions which is an indication of sufficiently stable trapping. We now added a sentence and a reference to the Discussion:

The acoustic radiation forces proportional to the gradient of the acoustic potential (Bruus, 2012), lead to a sufficient trap-stiffness to confine our investigated samples on the order of or smaller than $\lambda/4$. Importantly, between rotations and during the acquisition times of 8 s to 23 s to capture a single angle volume of a field of view ranging from $0.67 \text{ mm} \times 0.67 \text{ mm}$ to $0.76 \text{ mm} \times 3.78 \text{ mm}$, the sample could be kept stable by acoustic trapping, i.e., we did not observe any motion artifacts.

The sample does move slightly in all directions as a result of changing the settings between re-orientations and hence the pressure landscape, and these changes are corrected for, when needed, before starting the OCT scan. Further, if desired, the trap-stiffness can be improved by choosing higher trapping frequencies, and one could modify the chamber for additional confinement in y -direction if desired. We have added information about this in the Discussion:

(...) Reorientation leads to rotation around the major axis, but additionally there can also be a tilt due to asymmetry in the mass distribution and due to non-uniform acoustic forces in the chamber. The change in the acoustic trapping landscape to reorient the sample can also lead to small translations in any direction before the object again is stably trapped in the new local pressure minima. The translations and rotations are in general reversible when reverting the acoustic settings and small orientation changes, lead to small translations. Along the y -axis in the chamber, where we do not propagate acoustic waves, the sample is confined by the transverse component of the acoustic radiation forces from the (non-ideal 'plane') standing waves in the orthogonal directions (Woodside et al., 1997), and during rotations we generally observe larger drifts than in x - and z -direction where the trap-stiffness is higher. See Supplementary Methods for more details. The change in position and tilt is dealt with by our reconstruction algorithm and does not impose a serious restriction. For time-efficiency in imaging, we adjust the field of view between re-orientations to permit as small imaging volume as possible, and for the few significant translations along z -axis, we adjust the z -position of the sample stage.

One could alternatively capture a large FOV at the expense of a longer acquisition time. We now added more details about the XYZ drifts in the SI on p.15-16:

Each new orientation is a stable trapping position, but between each orientation (while rotating) the sample undergoes small translations. In general the drifts along the y -axis are larger where the trap-stiffness is lower. After the step-wise reorientation of about 90° in Supplementary Movie 2, the drift in both x - and y -direction is around $30 \mu\text{m}$ after adjusting the phase. After the rotation induced by adjusting the voltage, the sample drift along y -axis is in total $280 \mu\text{m}$ while the drift along the x -axis is around $20 \mu\text{m}$. Fine-tuning of the parameters of acoustic actuation, e.g., by limiting how much the voltage is adjusted, might reduce the drifts, and one could potentially also propagate acoustic waves in this third direction along the y -axis to improve the confinement. This could also open up for rotation of less asymmetric samples along two orthogonal object axes as demonstrated in our previous work (Løvmo et al., 2022).

Regarding quantification of the axial stability, we think this is an interesting question. We calculated the phase difference between adjacent A-lines at the same depth (Leitgeb et al., 2003, 2014). As illustrated in Fig. 1(a), we extracted intensity and phase information from the OCT spectra of a 3 dpf zebrafish. Based on the intensity image, a mask for the zebrafish was generated and applied to the phase difference image. The histogram of the masked region of the phase difference image is shown in Fig. 1(b). The phase difference within the zebrafish in this B-scan has a standard deviation of 1.004 radians and an average value of 0.003 radians. According to formula $\Delta z = \lambda_0 \cdot \frac{\Delta\phi}{4\pi}$, the acoustically trapped zebrafish had an axial phase stability of ± 67.5 nm and an average axial drift of 0.2 nm over the interval of 50 μ s (A-line rate: 20 kHz). The axial phase stability of the acoustically trapped zebrafish is comparable to that of a reference zebrafish embedded in Phytigel (± 63.6 nm to 81.4 nm) imaged using the same OCM system. This indicates that the measured axial phase instability originates mainly from the OCM system instead of the acoustic trapping. The trapped zebrafish had a small axial drift in the range of -2 nm to 7.6 nm over the interval of 50 μ s (as shown in Fig. 1(c)). According to the data of the reference zebrafish embedded in Phytigel, the axial drift had a positive systematic offset due to the scanners), and the drift sign varied, which implies the trapped zebrafish fluctuated within a small range. This small fluctuation did not affect the ULTIMA-OCT acquisition and reconstruction. After one volume acquisition, the trapped zebrafish remained in focus. We have now included this information in the SI on p. 8. as Supplementary Fig. 8:

and we have included a summary of these results in the Discussion which now reads:

The axial phase stability of the ULTIMA-OCT was characterized (as shown in Supplementary Fig. 7). By comparing with the data of a zebrafish embedded in Phytigel, it is found that the ULTIMA-OCT axial phase instability originated mainly from the OCM system rather than from acoustic trapping. There was a small axial fluctuation (within 10 nm) of the acoustically trapped zebrafish, but it did not affect the image acquisition and reconstruction. This makes the adaption of the acoustic chamber to functional OCT such as OCT angiography, Doppler OCT, spectroscopic OCT, and optical coherence elastography (Drexler and Fujimoto, 2008) possible to provide further biological information.

Reviewer Point P 2.4 — Could the authors evaluate the viability of their samples (spheroids and/or organoids) inside the chamber? Could the mechanical forces applied to the sample can affect differentiation and growth of the sample? Is the mechanical force asymmetric at the scale of the sample, and could it induce differential growth of different part of the sample? I am also a bit concern by the use of cyanoacrylate and nail polish to seal the glass plates for live imaging, which are known to affect the cells. Possibly, because the chamber volume is quite large, it may be neglectable. . . but I would rather see direct evidence that some samples could stay alive for several hours/days within the chamber.

Reply: As the reviewer points out, it is important to include assessments of the bio-viability in the next step, when addressing a specific biomedical problem (e.g., when studying the possible impact of mechanical/acoustic forces on differentiation and growth of the sample). We need to accommodate these amenities, but this should be straightforward, since a multitude of solutions for this have been published in the literature, as we refer to in the text below. As others have verified bio-compatibility at similar or higher pressures than we use for both cell models and zebrafish, so we do not expect effects on differentiation at these values.

Figure 1: The ULTIMA-OCT axial phase stability. a. The intensity and phase difference image of an example B-scan. Based on the intensity image, a mask was extracted and applied to the phase difference image to obtain the masked phase difference image. b. The phase difference histogram of the masked region of the phase difference image (std: standard deviation). c. The axial stability of the acoustically trapped zebrafish at different reorientation angles (blue square: average drift, error bar: standard deviation).

We share the opinion, that the effects of the used coatings might be low for a limited time period due to the large liquid volume around the sample. Bio-compatible glues and coatings are available and are one of the modifications we plan to include in future devices, alongside temperature control and micro-fluidics to allow for media exchange. In this study, we aimed to show a proof-of-principle of both the acoustic strategy in combination with OCT and collect data to develop the reconstruction algorithm, and a modification of the chamber and thorough testing of the biocompatibility is a planned collaboration as a next step. We modified the Discussion to include your comment regarding the glues and materials used and to make the expected bio-compatibility more clear. It now reads:

A multitude of similar acoustic trapping platforms (Bazou et al., 2011; Wiklund, 2012; Chen et al., 2021) have verified the biocompatibility of acoustic trapping, and our measured

pressures are below the limits of which adverse bio-effects and cavitation are expected. However, we will need to assess the effects of our specific acoustic trapping platform for long-term experiments on the targeted live samples in future experiments. The large sample volume in our chamber minimizes the effects of a potential harmful influence of the materials and glues used in this prototype and as bio-compatible glues and coatings are available, this does not pose a future concern. Details on how our acoustic chip can be modified to make it suitable for long-term monitoring of life sample with biocompatibility testing can be found in the Supplementary Methods.

Further notes regarding our measured pressures and expected biocompatibility can be found in SI on p. 11-12.

Questions and minor comments:

Reviewer Point P 2.5 — Introduction l 38-39: ‘Matrigel is complex and variable which gives rise to a certain irreproducibility’: Do the authors have a reference to support this claim? I am also wondering why they are focusing only on Matrigel, while there are many other potential gels that could be used? Do they know and can they discuss why such a focus (apparently in the community) was made on Matrigel?

Reply: Yes, indeed, we have taken this from the literature. The first lines in the abstract of the cited reference (Aisenbrey and Murphy, Nat. Rev. Mater., 2020) reads as follows: "Matrigel, a basement-membrane matrix extracted from Engelbreth–Holm–Swarm mousesarcomas, has been used for more than four decades for a myriad of cell culture applications. However, Matrigel is limited in its applicability to cellular biology, therapeutic cell manufacturing and drug discovery owing to its complex, ill-defined and variable composition. Variations in the mechanical and biochemical properties within a single batch of Matrigel — and between batches — have led to uncertainty in cell culture experiments and a lack of reproducibility."

Concerning the focus on Matrigel: We were under the impression that this was the most widely used gel by far, but we are no experts in this. Besides, we are aiming at contact-free manipulation, without any gel.

Reviewer Point P 2.6 — The authors describe the advantage of being far from the glass coverslip. However, in the current configuration, they need to cross about 1 cm of solution, which seems quite important. Do the authors need to compensate physically for dispersion? Would it be possible to reduce the size of the chamber to have the sample closer to the coverslip (e.g. to use higher NA objectives)

Reply: As stated in Methods, digital dispersion compensation was applied in the standard OCT preprocessing steps. We agree that it is of value to implement and try physical dispersion compensation in future work.

Future devices, with square cross-section could lower this distance and allow for the use of objectives with higher NA. Distance from coverslip as small as $\lambda/4$. We have modified the corresponding sentence which now reads:

Our future work will include investigations of simplified chamber designs with a square cross-section and two orthogonal transducers which would open up for manipulation closer to the bottom coverslip if desired for the targeted imaging setup.

Reviewer Point P 2.7 — How do the mechanical parameters, in particular viscosity, of the fluid influence the trapping? Often, Spheroids and organoids can secrete some molecules which can progressively change the local viscosity. Would it affect the trapping conditions and efficacy?

Reply: Probably the secreted molecules of a single spheroid in such a large volume is negligible. With future media exchange for live cell imaging, this effect is further reduced. Viscosity plays a role mainly for the spinning torque (also called the viscous torque), where a higher viscosity is correlated with a larger torque. The speed of sound and hence the resonance frequency depends on the density, compressibility and temperature of the liquid. We found a shift in the resonance frequency of 5 kHz between trapping in water and PBS. The viscosity difference between these liquids is around $0.13 \text{ mPa} \times \text{s}$. We added information about trapping in different liquids in the SI on p. 18 which now reads:

The resonance mode in the assembled chamber can be excited at the same frequency if the temperature is stable within a few degrees. If one performs trapping at a different temperature or in a different liquid, as for instance 1X PBS or cell media, one can quickly measure the electrical impedance again before the experiment. When changing from water to 1X PBS, we measured a positive shift in the resonance frequency of 5 kHz.

Reviewer Point P 2.8 — L 258: Could the authors describe here why a second line of Zebrafish is used, and why this one in particular? Is there a reason why this particular line is used rather than more typical (as far as I know) Nacre or Crystal phenotypes?

Reply: Thanks for proposing the crystal phenotype. We were interested in the *Mitfa*^{b692/b692}/*ednrb1*^{b140/b140} line and frequently used it in our previous studies because it has reduced melanophores and iridophores, which allows deeper penetration and better anatomy visualization. In this work, we would like to compare the differences (e.g. reconstruction algorithm capability, and attenuation map) between this phenotype and the wild-type. The eye of the *Mitfa*^{b692/b692}/*ednrb1*^{b140/b140} line is still dark compared with the crystal phenotype which is better for showing the advantages of ULTIMA-OCT. As far as we know, nacre is another name for the *Mitfa* mutation, and the crystal phenotype requires 3 homozygous mutations (nacre, alb, and roy mutation) as stated in the reference (Antinucci, P. *et al.*, A crystal-clear zebrafish for in vivo imaging. Scientific reports, 2016.). Our collaborator is also trying to develop the crystal phenotype. Once we have access to it, we would also like to image them using ULTIMA-OCT. We thus added the information in the methods/samples section:

*...and double mutant transparent *Mitfa*^{b692/b692}/*ednrb1*^{b140/b140} fish with reduced melanophores and iridophores (the second zebrafish line enabled deeper penetration and better anatomy visualization and was used in this work for comparison with the wild-type). (...)*

Reviewer Point P 2.9 — Reconstruction L.326: Could the authors provide references for the TV and Tikhonov regularization. I must admit that I am not an expert of such reconstruction, but I am missing the rationale for using such regularization? Are they completely standard, or could the authors provide reconstructions without such regularization, or with different ones?

Reply: The inverse problem we are trying to solve is ill-posed and therefore we used regularization to make the reconstruction process more stable. Both Tikhonov and total variation regularization are standard procedures that have been studied extensively in the literature, e.g. see (Rudin et al., 1992) for total variation (TV). Particularly the retrieval of the motion and the refractive index requires regularization for the solution to be unique. In the first stage of the multi-scale reconstruction, we extract the motion by running the proposed optimization algorithm with strong TV regularization on the reflectivity R . If the TV regularization is omitted at this stage, the solution is not unique, which results in a very bad estimate of the parameters (e.g. multiple non-aligned volumes are visible in the reconstruction). Intuitively, a correct alignment between volumes leads to reconstructions of R with a low TV norm (a wrong alignment leads to e.g. multiple boundaries in the solution, which gets penalized by the TV regularization).

Reviewer Point P 2.10 — Movie 4: could the authors explain what is the “filament” that seems to be under the fish? Is it intentionally added? Is it an artefact? If yes, is it common to have similar artefacts?

Reply: This fiber is a dust-particle, either from the solution or chamber or from the fixation process of the fish. We added this as a note in the Results which reads:

Similar artifacts were also identified in melanoma spheroids, where cells with high melanin levels limited the penetration depth due to high absorption. These shadow artifacts from a single viewing angle were also clearly revealed in the 3D rendering (see Supplementary Movie 4. Note that the structure below the sample is not an artifact, but an accidental contamination.)

Reviewer Point P 2.11 — Discussion L429 and also related to the first general question: It is not clear to me why the organoids have to be in suspension or in a gel droplet? Why is it complicated to have organoids in suspension? What does bring the gel droplet? Isn't it contradictory with all the efforts the authors made to design an acoustic trap to avoid adding gel around the organoid? Then, I am wondering if organoids can be a realistic application for this system? Could the authors add more comments on these points in the manuscript?

Reply: We would need the organoids in a suspension, and not in a gel to be able to acoustically manipulate the samples. Most organoids require gels to grow, and the procedures to form and mature organoids in matrigel are time-consuming and costly, as well as leading to the undesirable effects on the cells mentioned in the introduction. To remove the gel with a matrigel-removal is another step one has to perform to acoustically image the sample, and usually the organoids are put back into a gel after imaging for further maturation. Recent developments towards scaffold-free growth of many types of organoids are very promising. Some organoids are more dependent on a scaffold than others, in which case an organoid in a gel droplet could be imaged, also to compare and reveal effects of the gel. We have modified the paragraph in question and hope to have made this more clear:

In order to handle organoids in our platform, we would need the organoids in suspension and this will be a focus in future work. It has already been demonstrated that acoustofluidics can play a useful role in the trapping and merging of organoids (Ao et al., 2021), where organoids were formed in suspension or in Matrigel followed by Matrigel removal before acoustic manipulation and further growth in gel. Recent efforts has led to development of alternative scaffold-free growth of several types of organoids in suspension (Price et al., 2022; Weng et al., 2023). Organoid growth mediated by ultrasound may also be possible, similar to growing cancer spheroids under acoustic trapping (Chen et al., 2016; Olofsson et al., 2018; Jeger-Madiot et al., 2021).

Reviewer Point P 2.12 — Discussion L.512: In their work, the authors seem to use a 10X 0.3 NA objective with a depth of field around 10 μm , while they are imaging samples of several hundreds of microns. I understand that the multi-illumination helps, but how is it still possible to image at 100 μm depth at such resolution? Do they experience non-uniform lateral resolution in depth? Do they need to implement solutions such as ISAM and/or computed AO algorithms to compensate the resolution loss?

Reply: The reviewer is correct that we used 10 \times 0.3 NA objectives (CFI Plan Fluor 10 \times , Nikon) in this work, and the theoretical depth of field (DOF) of this objective should be around 10 μm if the objective aperture is filled properly. However, we utilized a reflective collimator (RC04APC-P01, Thorlabs) with a 3.9 mm (smaller than the objective entrance aperture) beam output to direct the OCM beam to the system. As a result, the OCM beam did not completely fill the objective aperture, leading to a larger DOF in practice.

One definition of DOF is twice the Rayleigh length. The corresponding measured DOF is around 153 μm in air (as shown in Fig. 2a) and 211.1 μm in sample (refractive index: 1.38). Because here the lateral resolution degraded from 3.4 μm to 3.9 μm over time, the measured DOF is a reference value. According to another definition of DOF by the full-width at half-maximum (FWHM) power (Drexler and Fujimoto, 2008), the measured DOF is approximately 248 μm in air (as shown in Fig. 2b) and 342.2 μm in sample.

Therefore, the DOF in this work is much larger than 10 μm . As the measured DOFs were still smaller than the zebrafish larvae, the lateral resolution degradation outside the confocal range did influence the reconstruction. Therefore, we agree it will be valuable to investigate the ISAM and computed AO algorithms proposed by the reviewer in future work, which can improve the reconstruction accuracy.

We have now added the DOF figure in the SI on p. 7 as Supplementary Fig. 7, and included all the component information and the measured DOFs in the Methods section which now reads:

The OCM lateral resolution was around 3.4 μm , and the depth of field (defined by twice the Rayleigh length, see Supplementary Fig. 6) was approximately 153 μm in air and 211.1 μm in tissue. Through reflective collimator (RC04APC-P01, Thorlabs) 1, the OCM beam was directed to the system and then divided by a beam splitter (70:30 (R: T), BS065, Thorlabs) to the sample arm and reference arm, respectively. The OCM laser was focused on the sample and the reference mirror using objectives (CFI Plan Fluor 10 \times , Nikon), and the laser power on the sample was around 1.53 mW. Reflective collimator (RC04APC-P01, Thorlabs) 2...

Figure 2: The measured depth of field (DOF) of the OCM system. a. The measured DOF defined by twice the Rayleigh length, $R_{x,min}$: minimal lateral resolution. b. The measured DOF defined by the full-width at half-maximum (FWHM) power, Z_s : the sample's position in the z-axis.

and in the Discussions which reads:

In this work, the OCM beam did not fully fill the objective aperture. This resulted in a larger depth of field (as shown in Supplementary Fig. 6), which worked for zebrafish larvae. However, to achieve higher reconstruction accuracy for large samples, interferometric synthetic aperture microscopy (Ralston et al., 2007) or other computational aberration correction (Kumar et al., 2013; Liu et al., 2017) could be investigated in the future.

Reviewer Point P 2.13 — Supplementary figure 5: Scale bars are missing

Reply: Thanks for spotting this. We added scale bars to the corresponding supplementary figure.

Reviewer Point P 2.14 — Could the authors add the reference of the microscope objective used? I had to guess it from the animation in video 1 to estimate the depth of field..

Reply: The objectives used in the setup are 'CFI Plan Fluor 10 \times , Nikon' (https://www.microscope.healthcare.nikon.com/en_EU/products/optics/selector/comparison/-1825) bought from the EO Edmund (Germany) (<https://www.edmundoptics.de/p/10x-objective-nikon-cfi-plan-fluor/30637>). This information has been added to the main text. The objective used in the animation for illustration is now modified to avoid confusion.

Ultrasound-Induced Reorientation for Multi-Angle Optical Coherence Tomography

Mia Kvåle Løvmo,^{1,*} Shiyu Deng,^{2,*} Simon Moser,^{1,*} Rainer Leitgeb,² Wolfgang Drexler,² and Monika Ritsch-Marte^{1,†}

¹*Institute of Biomedical Physics, Medical University of Innsbruck, Innsbruck, Austria*

²*Center for Medical Physics and Biomedical Engineering, Medical University of Vienna, Vienna, Austria*

(Dated: December 12, 2023)

Organoid and spheroid technology and model organism research provide valuable insights into developmental biology and oncology. Optical coherence tomography (OCT) is a label-free technique that has emerged as an excellent tool for monitoring the structure and function of these samples. However, mature organoids are often too opaque for OCT. Access to multi-angle views is highly desirable to overcome this limitation, preferably with non-contact sample handling. To fulfil these requirements, we present ULtrasound-Induced reorientation for Multi-Angle-OCT (ULTIMA-OCT), which employs a 3D-printed acoustic trap inserted into a spectral-domain OCT imaging system, to levitate and reorient zebrafish larvae and tumor spheroids in a controlled and reproducible manner. Our specifically-developed model-based algorithm performs a physically consistent fusion of the multi-angle data. We demonstrate enhanced penetration depth in the joint 3D-recovery of reflectivity, attenuation, refractive index, and position registration for zebrafish larvae. ULTIMA-OCT is an enabling tool for volumetric imaging with great potential for further applications.

Keywords: Optical coherence tomography, acoustic trapping, scattering tissue, tomographic reconstruction, multi-angle imaging, organoid research, zebrafish larvae, label-free, 3D refractive index

INTRODUCTION

A steep increase in organoid and spheroid research could be witnessed in recent years, providing vital insights into developmental biology and oncology. A strong motivation for this is the potential of organoids and cancer spheroids to reduce animal experimentation to some extent [1]. Spheroids can be grown with the support of an extracellular matrix or scaffold-free [2]. Organoids are usually grown in Matrigel, which is derived from the secretion of a type of mouse sarcoma cells. Matrigel is complex and variable which gives rise to a certain irreproducibility. Moreover, it is known that the matrix scaffolds have a mechanical impact which is often not well understood or characterized [3]. Therefore, in the past few years considerable effort has been directed towards Matrigel-free organoid growth [4].

In response to such problems, contact-free levitation of samples has been sought. On the single-cell level, it is possible to use holographic optical tweezers to induce rotations for tomographic studies [5, 6], but for cell-clusters reaching mm-size, optical trapping would lead to over-heating due to the intensities needed to counteract the growing weight [7], even in the most favourable case of counter-propagating optical beams creating a trapping region between two laser spots,

as in the macro-tweezers system [8], [9]. To tackle larger biological samples, ultrasound techniques for levitation and actuated handling have been developed: Standing bulk acoustic waves (BAW) operating at (sub-)MHz frequencies push biological cells into low-pressure regions (planes, lines or spots depending on the number of orthogonal standing waves) [10–12], and by modulation of the acoustic waves, objects can be transiently or continuously rotated [13–17]. Surface acoustic waves (SAW) can also be used, e.g., to create acoustofluidic rotational tweezers for morphological phenotyping of zebrafish larvae [18]. Streaming vortices generated by acoustically induced oscillations of bubbles or solid structures have been applied to rotate zebrafish embryos [19], single cells, pollen-grains and nematodes [20–23]. For compatibility with scanning-based imaging modalities, however, BAW operation is favorable, as it supports tilting of the sample into various stationary (non-rotational) orientations. Bulk waves also simplify device up-scaling for manipulation of large samples far from the chamber boundaries. Scaffold-free confinement in the center of the chamber represents a big advantage of acoustic levitation, since it makes the system more open to perform various assays, such as irradiating parts of the sample by light, adding chemicals and pharmaceuticals by micro-fluidics, or mechanical probing with tips.

Optical Coherence Tomography (OCT) [24–26], a technique based on low coherence interferometry, can reconstruct micrometer sample morphology from the backscattered light with high imaging speed (MHz A-scan rate) and has been widely employed for (bio)medical applications. Optical Coherence Microscopy (OCM), which uses higher numerical

* contributed equally

† Correspondence email address: monika.ritsch-marte@im-med.ac.at

Figure 1. **Schematics and workflow of ULTIMA-OCT**. **a** depicts the fluid-filled acoustic chamber in which the sample is levitated and reoriented by means of acoustic actuation. The specimen is rotated in a **step-wise manner** into several stable trapping positions **b**, and optical coherence microscopy (OCM) imaging is performed in each of them **c**. The acquired OCM data is **post-processed** **d** and fed into a multiscale optimization algorithm **e**, which performs fusion of the images and outputs 3D reconstructions **f** of reflectivity R , attenuation α and refractive index (RI) contrast Δn . In **g** an exemplary collection of samples is shown, where ULTIMA-OCT can be applied.

aperture (NA), can achieve high lateral resolution but usually is limited in penetration depth. OCT/OCM has emerged as a valuable imaging modality for **living tissues and model organisms** [27–29] and **more recently for organoids and spheroids** [30–32], providing high-resolution information on the internal structural organization inside the organoids non-invasively and **label-free**.

Nevertheless, mature specimens often become optically dense, intractable not only for Optical Diffraction Tomography (ODT) [33, 34] but also for OCT, leading to shadows and limited tissue morphology information. Shadow removal algorithms have been developed for OCT images of the optic nerve head [35–37]. However, removing the shadows cast by high-attenuation structures like the eye of a wild-type zebrafish larva remains challenging, because the OCT incident light

can be fully occluded. 3D optical coherence refraction tomography [38] **compensates** for this issue by controlling the incident beam angle and position using a parabolic mirror, but **this** was limited to $\pm 75^\circ$ angular orientation and needed to immobilize samples like zebrafish embryos in agarose gel.

In this work, we present an easy-to-use solution overcoming the above explained limitations and problems encountered when imaging organoids, spheroids or developing organisms: ULTIMA-OCT uses a small add-on microfluidic chamber with tunable **bulk** acoustic waves to stably and reproducibly rotate the **sample** into several orientations. This enables OCT imaging from different viewing angles in the full range of 360° around the sample’s major axis, which makes 3D tomographic

reconstruction feasible also for optically dense samples. The immobilization of the sample is contact-free and does not involve any rotating mechanical parts, nor any elements obstructing optical imaging or introducing optical aberrations.

The price to pay in this approach is the fact that one cannot precisely choose the exact viewing angles, since the stable trapping positions and orientations in the acoustic force fields to some extent also depend on the unknown sample itself [11]. However, we provide a generally applicable solution, i.e., a model-based algorithm which can deal with the added complexity of tomographic reconstruction with viewing angles that are not precisely known *a priori*.

RESULTS

Working principle of ULTIMA-OCT

The workflow of ULTIMA-OCT and its ingredients are explained schematically in Fig. 1, and a schematic animation of the data acquisition procedure can be found in Supplementary Movie 1. Acoustic radiation forces are used to levitate the sample and to induce transient rotations in a fluid chamber. Each new orientation is a stable trapping position, and we perform OCM scanning of the sample at a desired number of orientation angles. The 3D OCM data is then processed by a model-based algorithm to extract the underlying reflectivity map, while also yielding information about the attenuation and refractive index (RI) contrast maps as well as the position and orientation parameters of the trapped sample.

Acoustic actuation

The acoustic manipulation chamber consists of a 3D printed frame with a symmetric octagon cross-section with 4 piezo-electric plate transducers and 4 reflectors around its sides (Fig. 2 and Supplementary Fig. 1). We apply a sinusoidal signal to the transducers to propagate BAWs in 4 directions in the liquid-filled chamber and generate acoustic standing waves in each direction upon reflection. The standing waves have pressure nodes every $\lambda/2$ (≈ 1 mm around 600 kHz) along each propagation direction in the fluid chamber. The front and back of the octagon frame is sealed with a cover-slip and an aluminium plate respectively, and imaging is performed through the bottom cover-slip that also acts as the reflector for the acoustic waves from the top-transducer. Our current device is developed to accommodate a range of sample sizes and shapes and we have demonstrated this by manipulating samples, from

highly asymmetric mm-sized 3-5 days post fertilization (dpf) zebrafish embryos to less asymmetrically shaped sub-mm-sized melanoma spheroids. We rely on the resonant enhancement of the waves in the fluid chamber to get sufficient force to levitate our targeted biological samples. Levitation by the top transducer was achieved at a minimum peak-to-peak driving voltage of 20 V, corresponding to a maximum pressure amplitude of 80 kPa (see Methods and Supplementary Methods for details on used chamber dimensions and acoustic resonances).

To characterize and optimize our contact-less trapping platform for reorientation and multi-angle image acquisition, we used fixated 3 dpf zebrafish embryos, as they are readily available samples that are perfectly suited to demonstrate the benefits of our approach. To observe the zebrafish while tuning the acoustic settings for stable reorientation, we used an inverted microscope with oblique illumination through the front cover-glass of the chamber, acquiring dark-field images (Fig. 2c and details in Supplementary Methods). With the acoustic radiation forces [39–42] from the top transducer, we levitate the sample against gravity, in one of the nodal planes in the center region of the chamber where all 4 acoustic waves intersect. With the additional radiation forces from one side-transducer, we align the sample with its major axis to the length of the chamber (y -axis in Fig. 2).

By changing the voltage on the transducers, and hence the relative magnitudes of the acoustic radiation forces in each direction, we generate an acoustic restoring torque [11, 43, 44] acting on the sample. The torque direction is perpendicular to the acoustic propagation directions, hence parallel to the y -axis, and the direction of rotation is in the xz -plane (Fig. 2). With a dominating top transducer, the sample is aligned with its minor axis to the steepest trap-stiffness in z -direction (90° in Fig. 2c). When we increase the amplitude of one of the side-transducers step by step, we rotate the pressure landscape and the sample is rotated in a step-wise fashion until the sample is aligned with its minor axis to the now dominating forces from the side-transducer. We alternate between increasing and decreasing the amplitudes between pairs of transducers in a sequence, to rotate the sample 360° about its major axis, while ensuring levitation (top transducer voltage is tuned, but never zero).

Moreover, we found that for sufficiently asymmetric samples one can precisely control the orientation in a more efficient way, by exciting two overlapping orthogonal modes at exactly the same frequency in the chamber (by e.g., the top transducer and the orthogonal side-transducer S3 in Fig. 2) and adjusting the relative amplitude and phase (see Supplementary Methods for details on acoustic actuation), similar to [13] but

Figure 2. **Acoustic actuation for multi-angle imaging.** **a** Illustration of acoustic manipulation of levitated zebrafish embryo (not to scale). By coupling bulk acoustic waves into the fluid-filled chamber from multiple directions, acoustic standing waves (green) are generated upon reflection, to levitate the sample and induce transient rotations for optical imaging (red beam), e.g. for multi-angle high-speed OCM through the bottom cover-glass of the 3D printed octagon frame (black). The direction of rotation in the xz -planes is indicated (blue arrow). **b** Assembled octagon chamber with levitated zebrafish embryo (inside stippled red circle), scale bar: 5 mm. **c** The optimization of the acoustic actuation can be carried out on an inverted microscope with optical image acquisition. As an example darkfield (oblique illumination) images of a wild-type 3 dpf zebrafish embryo are shown here, for a selection of 8 chosen angles of acoustic reorientation, scale bar: 600 μm .

223 with additional levitation in our upright (not horizontal) chamber. In Fig. 2c we show dark-field images (2
 224 stitched tiles per image) from only 8 different orientations of a 3 dpf zebrafish, but we can reorient this sample
 225 with a much finer step-size, see Supplementary Movie 2 showing reorientation by two transducers of a zebrafish
 226 embryo.
 227

228 To demonstrate our capability of extending the out-
 229 lined acoustic manipulation to other types of samples,
 230 we also trapped and reoriented melanoma spheroids
 231 (see Supplementary Fig.4). These samples were smaller
 232 and less asymmetric than the zebrafish embryos, but
 233 we could reorient them around their major axis by
 234 the same-frequency two-transducer actuation described
 235 above. For other settings, however, sustained rotation
 236 was induced, with a rotation direction that could be
 237 reversed by changing the relative phase between the two
 238 orthogonal transducers by 180°, as also demonstrated
 239 by others [14] (see Supplementary Movie 3). It has so
 240 far always been possible to avoid unwanted sustained
 241 rotation, and to just stably reorient the spheroid 360°
 242 around one axis. In each orientation the sample is held
 243 stably without any significant motion, see Supplemen-
 244 tary Movie 2 and 3. Our trapping platform can accom-
 245 modate a large range of sample sizes and shapes, but
 246 each new sample needs its own fine-tuning of the acous-
 247 tic settings, which is performed on the fly while directly
 248 observing the object.
 249

251 High-resolution OCT imaging

252 To verify the feasibility of ULTIMA-OCT, biolog-
 253 ical samples such as fixated zebrafish larvae and
 254 melanoma spheroids were tested. The OCM system

255 successfully captured the 3D data of the samples
 256 levitated and reoriented in the acoustic chamber.
 257 Fig. 3 shows OCM images of a 3 dpf less-pigmented
 258 *Mitfa*^{b692/b692}/*ednrb1*^{b140/b140} zebrafish and a melanoma
 259 spheroid, respectively, imaged from one direction.
 260

261 The cross-section images (locations are indicated
 262 by the yellow dashed lines in the *en face* images in
 263 Fig. 3a) exhibit distinct shadow artifacts, as seen in
 264 Fig. 3b. For the zebrafish embryo, the eye with high
 265 melanin content and the yolk with high-attenuating
 266 internal structures cast shadows (signal loss) on deeper
 267 morphological features (marked by yellow dashed boxes
 268 in the OCM cross-section images). Similar artifacts
 269 were also identified in melanoma spheroids, where cells
 270 with high melanin levels limited the penetration depth
 271 due to high absorption. These shadow artifacts from a
 272 single viewing angle were also clearly revealed in the
 273 3D rendering (see Supplementary Movie 4. Note that
 274 the structure below the sample is not an artifact, but
 275 an accidental contamination.)
 276

277 *En face* images obtained by average intensity projection
 278 or standard deviation projection display the combined
 279 signal from different sample depths. Naturally, the ar-
 280 tifacts caused by shadowing are not as obvious in this
 281 type of visualization. Fig. 4 demonstrates such OCM
 282 images, with less noticeable shadowing, of a 5 dpf less-
 283 pigmented zebrafish embryo obtained from eight viewing
 284 angles. Whole-body *en face* data were obtained from
 285 three angles (indicated by the sub-image frame colors).
 286 Zebrafish features such as eye, otolith, yolk, muscle, no-
 287 tochord, and fins were discerned clearly by the OCM
 288 setup. Complementary zebrafish features were visual-
 289 ized from individual angles, but darker regions were ob-

290 served in the images, depending on the reorientation angles.
 291

Figure 3. OCM limitations. **a** Average intensity projection for zebrafish embryo and standard deviation projection for melanoma spheroid, both in logarithmic scale (OCM *en face* images). **b** Cross-section images for zebrafish embryo and melanoma spheroid in logarithmic scale. Cross-section image positions are indicated by the yellow dash lines in the OCM *en face* images in **a**. Shadow artifacts are indicated by the yellow dashed boxes in the OCM cross-section images. Scale bars: 200 μm .

292 Reconstruction

292

293 The sample can be reoriented in a reproducible manner
 294 by acoustic actuation, but – in contrast to externally
 295 induced mechanical rotation of an object immobilized in
 296 a container – the exact orientation between the recorded
 297 OCM volumes is unknown a priori. Depending on the
 298 orientation, different parts of the sample are occluded
 299 due to attenuation by structures in the sample, which
 300 for a zebrafish embryo is especially pronounced for the
 301 eye and the yolk sac. Also, structures of different RI
 302 inside the sample cause a local delay or surge of the
 303 recorded A-scan [38, 45, 46]. This effect is especially
 304 visible as a delay for structures behind the lens portion
 305 of the eye.

306
 307 Due to the mentioned distortion, the OCM volumes
 308 belonging to different orientation angles are not simply
 309 related by a rigid body transform, but correspond to
 310 each other in a more complicated manner. Moreover,
 311 the shadowing artifacts hinder a reliable registration
 312 of the orientations of the different volumes. Therefore,
 313 we formulate the fusion as an inverse problem, where
 314 the OCM image formation is expressed as a physical

Figure 4. OCM *en face* images of acoustically reoriented 5 dpf zebrafish embryo. **a** OCM *en face* images of the head section and **b** full body images of a zebrafish embryo (mutation $Mitfa^{b692/b692}/ednrb1^{b140/b140}$). Frame colors indicate corresponding angles. Scale bars: 200 μm .

315 forward model. This approach grants us the flexibility
 316 to deal with these uncertainties by constraints and
 317 regularization. This includes total variation (TV) and
 318 Tikhonov regularization as well as positivity and object
 319 support constraints. We solve this inverse problem
 320 by means of a gradient-based optimization approach,
 321 whose optimization parameters consist in the underlying
 322 reflectivity map R as well as attenuation α , RI contrast
 323 Δn and motion parameters q (rotation parameterized
 324 by unit quaternions) and t (translation). For a detailed
 325 description, we refer to the Supplementary Methods.

326
 327 Fig. 5 and 6 show the reconstruction results of ULTIMA-
 328 OCT from the head section of 3 dpf zebrafish embryos,
 329 a wild-type and a less-pigmented mutation, respectively.
 330 In a comparison of the OCM volumes and the recon-
 331 struction of the reflectivity map R , both depicted in
 332 logarithmic scale, one clearly appreciates the benefit of
 333 the proposed approach. One can see that both speci-
 334 mens strongly attenuate the signal in the OCM volumes,
 335 whereas the reconstructed reflectivities no longer show
 336 attenuation or distortion artifacts. In the wild-type
 337 zebrafish, the scattering and absorptive structures con-

tained within the total attenuation α are present across the whole head section, whereas for the mutated specimen the eyes account for most of the attenuation. Vertebrate eye-lenses exhibit a graded-index (GRIN) profile, which increases towards the center. As the most prominent structures of RI map are those belonging to the eye-lenses, we employ regularization that promotes smoothness, whereas for the attenuation and reflectivity maps we use edge-preserving regularization. The values obtained for the RI using this method are consistent with values from the literature [47, 48]. The dataset of the wild-type zebrafish embryo consists of OCM recordings of 11 angles, which are roughly distributed between 0 and 360°, whereas for the less pigmented mutation zebrafish embryo, 10 angles were used. 3D visualization of reconstructions comparing single- and multi-view of the wild-type zebrafish embryo can be found in Supplementary Movie 5 and 6 with a 3D rendered object and a flythrough-animation, respectively.

DISCUSSION

ULTIMA-OCT imaging and tomographic reconstruction provide volumetric information on the sample with enhanced penetration depth. This is achieved by acoustic reorientation, without the need for any moving mechanical parts, and in a non-contact way without the need of a supporting scaffold. This makes the sample much more accessible for mechanical probing and facilitates unobstructed and undistorted imaging. We will now discuss some current difficulties and limitations as well as possible extensions of the approach in the future.

Ultrasound trapping offers several advantages for OCT imaging. The standard method to study samples such as live zebrafish larvae with OCT has been to manually position the larvae under anesthesia on a gel layer on a coverslip and image from one direction. The purpose of this gel layer is to lift the sample up to create sufficient distance from the coverslip, and the coverslip is usually tilted to get rid of the reflection from the bottom coverslip. As a result, parts of the big sample may be out of focus and requires focus stacking or image stitching. ULTIMA-OCT has an acoustic chamber that can levitate samples up to 1 cm away from the bottom and top chamber surfaces, preventing these issues. Manipulating samples far from reflecting surfaces might not be possible for alternative acoustic strategies utilizing SAW as such devices are difficult to scale up. Further, we are unaware of stable reorientation of levitated and strongly asymmetric samples such as zebrafish larvae in devices utilizing acoustically induced microstreaming from bubbles, solid structures or SAW, although demonstrated for the more axis-symmetric *C. elegans* [49].

To avoid motion artifacts in OCT imaging of live

zebrafish anaesthesia is necessary. We do not anticipate any issues in achieving stable trapping of live sedated fish, since behavioural response under the influence of optical tweezers has been successfully studied under similar conditions [50]. OCT imaging of wake fish embryos does not seem feasible, since the considerably stronger acoustic forces necessary to stall an actively swimming fish will have non-negligible bio-effects (such as heating and cavitation), besides the problem of motion artifacts that will be inevitable. The acoustic radiation forces proportional to the gradient of the acoustic potential [40], lead to a sufficient trap-stiffness to confine our investigated samples on the order of or smaller than $\lambda/4$. Importantly, between rotations and during the acquisition times of 8 s to 23 s to capture a single angle volume of a field of view ranging from 0.67 mm×0.67 mm to 0.76 mm×3.78 mm, the sample could be kept stable by acoustic trapping, i.e., we did not observe any motion artifacts.

The shape of the levitated specimen has an influence: For elongated and very asymmetrical samples such as the zebrafish embryos, we found that we can precisely control the orientation to basically any desired angle, as seen in Supplementary Movie 2. The induced rotation angle is proportional to the step size in the adjusted voltage or phase, and by choosing a smaller step size than the 1 V used in this example, we can achieve finer precision. Reorientation leads to rotation around the major axis, but additionally there can also be a tilt due to asymmetry in the mass distribution and due to non-uniform acoustic forces in the chamber. The change in the acoustic trapping landscape to reorient the sample can also lead to small translations in any direction before the object again is stably trapped in the new local pressure minima. The translations and rotations are in general reversible when reverting the acoustic settings and small orientation changes, lead to small translations. Along the y -axis in the chamber, where we do not propagate acoustic waves, the sample is confined by the transverse component of the acoustic radiation forces from the (non-ideal 'plane') standing waves in the orthogonal directions [51], and during rotations we generally observe larger drifts than in x - and z -direction where the trap-stiffness is higher. See Supplementary Methods for more details. The change in position and tilt is dealt with by our reconstruction algorithm and does not impose a serious restriction. For time-efficiency in imaging, we adjust the field of view between re-orientations to permit as small imaging volume as possible, and for the few significant translations along the z -axis, we adjust the z -position of the sample stage.

Since the targeted types of samples are never perfectly spherical, our manipulation strategy is also suitable for less asymmetric samples as shown in Supplementary Fig . 4 and Movie 3, where we successfully manipulated

Figure 5. **Reconstruction results for the head section of a 3 dpf wild-type zebrafish embryo.** In **a-d** the $(y-z)$, **e, f, h, i** the $(x-z)$ and **j-m** depict the $(x-y)$ sections of the reconstructions. The leftmost column (**a, e, j**) shows the sections of the recorded OCM volumes in logarithmic scale, whereas the adjacent column (**b, f, k**) shows the reconstructed reflectivity map R of the same sections in logarithmic scale. **d, i, m** show the slices of the reconstructed attenuation map α (in mm^{-1}), whereas **c, h, l** depict the sections through the reconstructed RI distribution n . In **g**, a 3D rendering of the reconstructed reflectivity map is shown, together with the planes shown in **a-f, h-m**. Scale bars: $100\ \mu\text{m}$.

451 melanoma spheroids. In order to handle organoids in our
 452 platform, we would need the organoids in suspension and
 453 this will be a focus in future work. It has already been
 454 demonstrated that acoustofluidics can play a useful role
 455 in the trapping and merging of organoids [52], where
 456 organoids were formed in suspension or in Matrigel
 457 followed by Matrigel removal before acoustic manipula-
 458 tion and further growth in gel. Recent efforts has led to
 459 development of alternative scaffold-free growth of several
 460 types of organoids in suspension [53, 54]. Organoid
 461 growth mediated by ultrasound may also be possible,

462 similar to growing cancer spheroids under acoustic trap-
 463 ping [55–57].

464 The range of trappable sample sizes depends on the
 465 design of the acoustic chamber. In previous studies,
 466 we found that we can control transient and sustained
 467 rotation of samples up to a thickness around $\lambda/2$
 468 [11, 17]. Our earlier platforms implementing acoustic
 469 trapping around 3 MHz were limited to manipulation
 470 of samples up to roughly 200 μm in size. These devices
 471 were restricted to trapping close to the bottom coverslip
 472 and the induced transient rotations were typically

Figure 6. **Reconstruction results for the head section of a 3 dpf zebrafish embryo of a mutation with reduced pigmentation.** In **a-d** the $(y-z)$, **e, f, h, i** the $(x-z)$ and **j-m** depict the $(x-y)$ sections of the reconstructions. The leftmost column (**a, e, j**) shows the sections of the recorded OCM volumes in logarithmic scale, whereas the adjacent column (**b, f, k**) shows the reconstructed reflectivity map R of the same sections in logarithmic scale. **d, i, m** show the slices of the reconstructed attenuation map α (in mm^{-1}), whereas **c, h, l** depict the sections through the reconstructed RI distribution n . In **g**, a 3D rendering of the reconstructed reflectivity map is shown, together with the planes shown in **a-f, h-m**. Scale bars: 100 μm .

473 limited to 90° . The current work provides a solution
 474 to these limitations and facilitates manipulation of
 475 mm-sized samples at 0.6 MHz and 2.5 mm wavelength
 476 in water. The symmetry of the current chamber allows
 477 for excitation of the same orthogonal resonance modes
 478 which facilitates 360° step-wise rotation with two
 479 transducers. For large samples relative to the trapping
 480 wavelength, the acoustic radiation forces scaling with
 481 the sample radius becomes more complicated [39, 41]
 482 than in the small particle limit [58]. However, we are
 483 not at the limits in driving voltage and we believe our

484 platform is also suitable for handling larger samples
 485 up to a thickness of above 1 mm. Concerning the
 486 lower bound for sample sizes, we have demonstrated
 487 trapping and reorientation of a cancer spheroid of
 488 about 400 μm thickness. The occurrence of sustained
 489 rotations depends on the sample shape and could be
 490 suppressed for our investigated samples, but the exact
 491 limits of size and shape need further investigation. If
 492 higher trap-stiffness is found necessary to manipulate
 493 significantly smaller sample sizes, one could either
 494 explore using a higher harmonic frequency in the same

device, or transducers of higher resonance frequency, which we elaborate on in the Supplementary Methods along with details on choice of transducers and size of the acoustic chamber.

A multitude of similar acoustic trapping platforms [18, 59, 60] have verified the biocompatibility of acoustic trapping, and our measured pressures are below the limits of which adverse bio-effects and cavitation are expected. However, we will need to assess the effects of our specific acoustic trapping platform for long-term experiments on the targeted live samples in future experiments. The large sample volume in our chamber minimizes the effects of a potential harmful influence of the materials and glues used in this prototype and as bio-compatible glues and coatings are available, this does not pose a future concern. Details on how our acoustic chip can be modified to make it suitable for long-term monitoring of life sample with biocompatibility testing can be found in the Supplementary Methods.

The targeted samples are addressable by OCM, with the penetration depth depending on the light source’s wavelength and the sample’s scattering and attenuation properties. Longer wavelengths allow for deeper penetration at the cost of lower resolution. High-scattering and high-attenuation structures can obstruct the incident beam, resulting in shallower penetration. Therefore, OCT imaging depth is limited to 1 mm to 2 mm for most biological samples, and in OCM it can even be less. To achieve high lateral resolution, the numerical aperture is increased in OCM. However, this can result in a decrease in the depth of field and lead to non-uniform lateral resolution at varying depths. Ultrasound imaging can reach deeper but has a poorer resolution. ULTIMA-OCT maintains the high resolution of OCT/OCM and compensates for shadow artifacts, thereby reconstructing the sample part beyond the limits of a single angle’s penetration. In this work, the OCM beam did not fully fill the objective aperture. This resulted in a larger depth of field (as shown in Supplementary Fig. 7), which worked for zebrafish larvae. However, to achieve higher reconstruction accuracy for large samples, interferometric synthetic aperture microscopy [61] or other computational aberration correction [62, 63] could be investigated in the future. The axial phase stability of the ULTIMA-OCT was characterized (as shown in Supplementary Fig. 8). By comparing with the data of a zebrafish embedded in Phytigel, it is found that the ULTIMA-OCT axial phase instability originated mainly from the OCM system rather than from acoustic trapping. There was a small axial fluctuation (within 10 nm) of the acoustically trapped zebrafish, but it did not affect the image acquisition and reconstruction. This makes the adaption of the acoustic chamber to functional OCT such as OCT angiography, Doppler OCT, spectroscopic OCT, and optical coherence

elastography [64] possible to provide further biological information. Our acoustic actuation strategy might also be combined with other imaging techniques such as fluorescence-, multi-photon-, photoacoustic microscopy, and Raman spectroscopy, also in conjunction with OCT in dual-/multi-modal systems for more applications at different organization-levels [64]. Our future work will also include investigations of simplified chamber designs with a square cross-section and two orthogonal transducers which would open up for manipulation closer to the bottom coverslip if desired for the targeted imaging setup.

The tomographic reconstruction of reflectivity, attenuation and RI using fewer viewing angles, performed in this work and explained in the Methods section below, uses fewer viewing angles compared to previous work on samples immobilized in gel [38, 46]. However, the greater uncertainty in the orientation of the sample in our case adds additional ambiguity, which makes the reconstruction process even more challenging. To achieve sufficient accuracy on the registered angles we made use of prior information in the reconstruction process. In the first stage of the algorithm, it is crucial to make heavy use of regularization on the reflectivity map R to deal with the unknown reorientation in-between volume recordings. To make this step efficient, coarser representations of the recorded data can be used to obtain a first guess of the reflectivity, RI, attenuation map and motion. After the motion is registered with sufficient accuracy, the high-resolution reconstruction can be initialized with the parameters estimated from the first stage. Additionally, the strength of the regularization (explained in the Methods section and in more detail in the Supplementary Methods) can be lifted in order to also record fine-grained structures and to make use of the full resolution in the OCT dataset.

Limitations of the presented reconstruction approach concern the attenuation and RI maps. In modeling the attenuation we assumed the scattering and absorption to be independent of the recording angle. While the attenuation map is a useful quantity for the fidelity of our model, the angular independence might not be given and therefore the reconstruction of α is limited in its informative value. As the number of orientations used in this work is small, the reliability of the RI map outside the eye regions may be restricted. Although the RI values of the lens portion of the eye can be estimated accurately, the reconstruction of the RI map is strongly dependent on the available angular coverage in the dataset and the specimen itself. As a change in the RI at a location manifests itself as a delay in the structures behind that location, distinct structures have to be visible in the OCT signal. Since the reconstructed maps for reflectivity, attenuation and RI are tightly linked physically, future work on the image fusion could entail utilizing a more refined physical forward model

in the form of a unified treatment of those quantities. Our strategy of multi-angle OCT could potentially also profit from a full wave-optical treatment of the light-matter interaction [65].

ULTIMA-OCT combines cutting edge modalities in acoustic actuation, OCT and model-based tomographic reconstruction. Our 3D printed low-cost chamber is a simple add-on to an OCT imaging platform permitting multi-angle acquisition of a large range of samples. The presented strategy of step-wise reorientation, registration and reconstruction can be applied to a wide range of microscopy techniques. OCT is particularly suitable to give insight into dynamics and morphological changes in live biological samples due to its label-free and non-invasive nature. Through genetic overlap with model organisms or human-derived *in vitro* models, long-term monitoring of these samples provide valuable insights into human disease. The advancement of patient-derived spheroid and organoid models holds promise for personalized medical diagnosis and treatment, and for acceleration of oncology studies, for instance, compared to animal models. Our acoustic chamber not only offers a solution for non-contact manipulation of these samples, but the multi-angle OCT acquisition combined with our reconstruction algorithm enables a detailed reconstruction of samples at an enhanced penetration depth. Consequently, ULTIMA-OCT permits 3D reconstruction of larger or more optically dense samples and we believe this technique holds great potential in biomedical research for quantitative analysis of developing structures to distinguish and quantify for instance volume and surface of internal structures and voids or necrotic regions and for tracing effects from drugs or genetic modifications.

METHODS

Acoustic manipulation chamber

The chamber frame with a symmetric octagon cross-section is 3D-printed (Original Prusa i3 MK3, Prusa Research, Czech Republic) in a polymer (PET-G, RS: 891-9309) with open front- and backside and with windows around the 8 sides for attaching 4 piezo-electric transducers and 4 reflectors. The 4 transducers are positioned on the top part of the chamber, as shown in Fig. 2b and Supplementary Fig. 1 (top-transducer and S1-S3 side-transducers) with the 4 reflectors on the opposite parallel side. For imaging compatibility through the bottom of the chamber, a 170 μm thick coverslip seals the bottom and acts as the reflector of the acoustic waves from the top-transducer. The remaining three reflectors (R1-R3 in Supplementary Fig. 1) are machined in aluminium or cut from a 170 μm coverslip. All 4 transducers are (8 mm \times 15 mm, 3 mm thick) plate transducers made of Lead Zirconate

Titanate (PZT) (Pz26, CTS Ferroperm, Denmark). With the aim of levitating samples of a size in the mm-range, we chose these transducers with a thickness resonant mode frequency of the bare transducer around 670 kHz (wavelength in water of about 2.2 mm). Resonantly enhanced bulk acoustic waves generate standing waves of sufficient force in our chamber to levitate and reorient our samples. The specific chamber height used here (19.2 mm) was found by an iterative approach of adjusting the chamber dimensions of the 3D printed frame based on simulations and characterizing the acoustic resonances by electrical impedance measurements and experiments. See Supplementary Fig. 2 for specific chamber dimensions and detailed information in Supplementary Methods.

The backside of the printed chamber frame is covered by an octagon-shaped aluminium plate, to seal the back of the chamber. To attach the parts to the printed chamber frame, we use cyanoacrylate glue followed by nail-polish to completely seal all remaining gaps. The bottom silver plated electrode of each piezo-plate is connected to the aluminium with silver paint (RS: 123-9911) for thermal and electrical connection (common ground). To electrically connect to the top- and the bottom electrodes on each transducer we use copper wires and silver-paint to the top electrode and aluminium plate respectively (Supplementary Fig. 1). We drive each transducer with a sinusoidal signal from waveform generators: two single output waveform generators (Agilent 33220A) to drive side-transducers S1 and S2 (at 590 kHz), and one dual output waveform generator (Keysight 33522B) to drive the top- and S3-transducer (at 600 kHz). Each signal is amplified by power amplifiers (EVAL-ADA4870, Analog Devices) and impedance matching transformers, see details in Supplementary Methods. To ensure levitation of our samples we operate the top-transducer above 20 V, and to reorient the samples we tune the voltages of the transducers in the range of 20 V to 35 V. This corresponds to maximum pressure amplitudes in the range of 80 kPa to 150 kPa measured in the anti-node with a hydrophone (NH0200, Precision Acoustics, United Kingdom) . Please note that voltage refers to peak-to-peak voltage throughout the paper.

Operation for multi-angle imaging

The assembled octagon chamber is placed in a 3D-printed sample holder (Supplementary Fig. 1) that fits on the inverted microscope stage, attached via an adapter above the objective in the imaging setup in the case of OCT. We tilt the holder 90° so the open chamber front faces upwards and rinse the chamber with a 0.1% Triton X-100 solution (Sigma-Aldrich) to make the chamber more hydrophilic, to limit bubble formation at surfaces when filling the chamber. We then fill the chamber with the liquid (distilled water, tap water

720 or 1X phosphate-buffered saline (PBS)), place the sample
 721 inside and seal the front with a coverslip. This front
 722 coverslip is kept in place by adhesion forces, and can
 723 easily be removed for sample exchange. The top trans-
 724 ducer is turned on and we tilt the chamber to levitate
 725 the sample in the middle region of the chamber. The
 726 holder is placed on the imaging stage, and we start the
 727 acoustic manipulation to reorient the sample and acquire
 728 images through the bottom coverslip of the chamber at
 729 each desired step. Darkfield image acquisition (see Sup-
 730 plementary Methods for setup details) was used before
 731 performing OCM to optimize the acoustic settings for
 732 stable reorientation. The acoustic rotation and OCM
 733 imaging procedure is illustrated in Supplementary Movie
 734 1 and Supplementary Fig. 3. The top transducer is
 735 "far away" from the trapped sample (about 7 mm to
 736 12 mm), but to further limit the back-reflection from
 737 the top transducer during OCM, we paint the bottom
 738 silver plated electrode black, which does not affect its
 739 acoustic performance.

740 Samples

741 Zebrafish (*Danio rerio*) embryo preparation.

742 In this work, we used zebrafish of the pigmented
 743 wild-type Tubingen strain (wild-type) and double
 744 mutant transparent *Mitfa*^{b692/b692}/*ednrb1*^{b140/b140} fish
 745 with reduced melanophores and iridophores (the second
 746 zebrafish line enabled deeper penetration and better
 747 anatomy visualization and was used in this work for
 748 comparison with the wild-type). After spawning, eggs
 749 were maintained in egg water at 28 °C under standard
 750 conditions for up to 3 and 5 dpf. After overnight
 751 fixation in 4% paraformaldehyde (PFA), embryos were
 752 washed with PBS and were stored at 4 °C until they
 753 were used for imaging.

755 Spheroid preparation.

756 The murine melanoma cell line B16-F10 was used to form spheroids by the hanging
 757 drop method. Cells were grown in Dulbecco's Modified
 758 Eagle Medium supplemented with 10% fetal bovine
 759 serum, 1% penicillin/streptomycin in a humidified in-
 760 cubator at 37 °C, 5% CO₂. When 80% cell confluency
 761 was reached, cells were trypsinized, and 1000 cells per
 762 25 µl media were placed as droplets onto a Petri dish lid,
 763 inverted, and returned as top of the Petri dish bottom
 764 part, which was filled with 15 ml PBS for humidity
 765 and incubated for 4 days. During this time, individual
 766 spheroids formed in each droplet. The spheroids were
 767 collected in microcentrifuge tubes, washed 3 times with
 768 PBS between centrifugation (300×g, 5 min), and fixed
 769 with 4% PFA at room temperature for 20 min before
 770 washing 3 times again with PBS. Spheroids were then
 771 stored at 4 °C until use.

Adaptation of OCT set-up

774 Figure 7. The schematic of the OCM system and the
 775 add-on acoustic chamber. RC: reflective collimator; M:
 776 mirror; BS: beam splitter; VNDf: variable neutral density
 777 filter; CM: concave mirror; O: objective; G_x : x galvanometer
 778 scanner; G_y : y galvanometer scanner.

774 The setup [66] (as illustrated in Fig. 7) used in this
 775 work is a spectral domain OCM system using a compact
 776 polarization-aligned three-superluminescent-diode laser
 777 source (EBD290002, EXALOS AG) [67]. The OCM
 778 laser source had a center wavelength of 845 nm and a
 779 wide bandwidth of 131 nm, resulting in a high axial
 780 resolution of approximately 3.7 µm in air, corresponding
 781 to 2.68 µm in tissue (with a RI of 1.38). The OCM
 782 lateral resolution was around 3.4 µm, and the depth
 783 of field (defined by twice the Rayleigh length, see
 784 Supplementary Fig. 7) was approximately 153 µm
 785 in air and 211.1 µm in tissue. Through reflective
 786 collimator (RC04APC-P01, Thorlabs)₁, the OCM
 787 beam was directed to the system and then divided
 788 by a beam splitter (70:30 (R: T), BS065, Thorlabs)
 789 to the sample arm and reference arm, respectively.
 790 The OCM laser was focused on the sample and the
 791 reference mirror using objectives (CFI Plan Fluor
 792 10×, Nikon, NA = 0.3), and the laser power on the
 793 sample was around 1.53 mW. Reflective collimator
 794 (RC04APC-P01, Thorlabs)₂ was connected with a
 795 homemade spectrometer [67, 68] to capture the inter-
 796 ferogram of backscattered light from the reference
 797 mirror and the sample. The acoustic chamber was
 798 mounted on a chamber holder and implemented in the
 799 OCM system using a 3D-printed adapter. Precise sam-
 800 ple positioning and focus adjustment were achieved
 801 using a three-axis translation stage (MAX313D/M, Thorlabs).

803 Volumetric OCM data was obtained by the raster scan-
 804 ning of a pair of galvanometer scanners (CTI6220H,

Cambridge Technology) inside a conjugated scanning stage. During imaging, a 20 kHz camera line scan rate of the OCM spectrometer was used, corresponding to a sensitivity of 104.7 dB. For zebrafish embryo imaging, a scanning step size of 1.68 μm was used for smaller field-of-view imaging (head region), and 2.52 μm was used for whole fish imaging. A scanning step size of 1.68 μm was employed for melanoma spheroid imaging.

After standard OCT preprocessing steps (background subtraction, resampling, digital dispersion compensation, fast Fourier transform, and logarithmic calculation), OCM raw binary data was converted to three-dimensional images [69]. *En face* images of the zebrafish embryos were obtained by average intensity projection, and the *en face* image of the melanoma spheroid was obtained by standard deviation projection using Fiji [70]. To create the OCM cross-section image of the zebrafish embryo, 7 B-scans were averaged consecutively. Similarly, the OCM cross-section image of the melanin spheroid was obtained by averaging 3 consecutive B-scans. 3D rendering of the volumetric data was achieved using Amira 3D (Thermo Fisher Scientific, version 2023.1.1).

Reconstruction algorithm

The dataset consisting of OCM volumes of the specimen at multiple different orientations serves as the starting point for the reconstruction algorithm. We follow a model-based approach to describe the observed OCM data by the interaction of a reflectivity R , attenuation α and RI contrast map Δn . Inspired by the works of [38, 46, 71], the detected OCM signal I is modeled line-wise by a layer-by-layer based propagation

$$I_i = R(\mathbf{r}_i) \cdot H(z_i) \cdot T(z_i) \cdot \exp(-2 \sum_{j=0}^i \alpha(\mathbf{r}_j) \Delta z_j) \quad (1)$$

$$z_{i+1} = z_i + \frac{\Delta z}{n_0 + \Delta n(\mathbf{r}_i)}. \quad (2)$$

Starting with \mathbf{r}_0 as the boundary conditions for the coordinates, the signal is traced through the specimen represented by R , α and Δn . T and H denote the confocal point spread function and sensitivity roll-off, respectively [72].

To extract the maps for R , α and Δn we formulate the reconstruction as an optimization problem. The error metric we aim to minimize is composed of data fidelity, total variation, and l_2 -norm, as well as positivity constraints on R , α and n . In addition to R , α and n , also \mathbf{q} and \mathbf{t} , rotation parameterised by quaternions and translations, represent optimization parameters. We solve the optimization problem jointly

with Stochastic Gradient Descent (SGD), where we first extract motion parameters and reconstructions on a low-resolution representation. Afterwards, the high-resolution reconstruction is initialized with the resulting obtained low-resolution quantities, and further refined iteratively to yield the final reconstruction. For a detailed explanation, we refer to Supplementary Fig. 6 and Supplementary Methods.

The numerical optimization was conducted in JAX [73] 0.4.13 using Python 3.11 on a workstation equipped with an Nvidia RTX 4090 GPU. To obtain the low-resolution reconstructions we ran the algorithm for 1000 iterations and used a 3x smaller resolution (134^3 and 150^3), whereas for the high-resolution reconstructions (400^3 and 450^3) 250 iterations were performed. An iteration represents a single pass through the entire dataset. The whole reconstruction process took approximately 20 min.

FUNDING

This work was funded in part by the Austrian Science Fund (FWF) SFB 10.55776/F68 Tomography across the scales, projects F6803-N36 (Multi-Modal Imaging) and F6806-N36 (Inverse Problems in Imaging of Trapped Particles), and in part by H2020-ICT-2018-20 project REAP with grant agreement ID 101016964, and the Joint Ph.D. Program Medical University of Vienna/NTU Singapore “Kooperation Singapur” (Grant No. SO10300010).

ACKNOWLEDGMENTS

We thank Nicole Schmitner (Institute of Molecular Biology, University of Innsbruck) for providing us with zebrafish embryos, describing the features and sample preparation, as well as for valuable discussions, Abigail J. Deloria (Center for Medical Physics and Biomedical Engineering, Medical University of Vienna), Agnes Csiszar, and Gergely Szakacs from the Center for Cancer Research, Medical University of Vienna for providing us with melanoma spheroids and the sample preparation description. We thank Gregor Thalhammer-Thurner (Institute of Biomedical Physics, Medical University of Innsbruck) for valuable discussions and for proof-reading. We would like to express our gratitude to Richard Haindl (Center for Medical Physics and Biomedical Engineering, Medical University of Vienna) for the OCM software development and the homemade spectrometer build. We thank M. Duell from the EXALOS AG for the OCM laser source. We would like to acknowledge the FWF doc.funds PhD program Image-Guided Diagnosis and Therapy (IGDT) at the Medical University of

Innsbruck in which MKL is enrolled.

DATA AVAILABILITY

The datasets supporting the findings in this study are available from the corresponding author upon reasonable request.

CODE AVAILABILITY

[The codes are available at https://github.com/simo343/ultimaoct.](https://github.com/simo343/ultimaoct)

SUPPLEMENTARY INFORMATION

The Supplementary Material for this article can be found online at: to be specified. It covers details

on acoustic chamber design and assembly including possible future extensions, as well as an example of a reoriented melanoma spheroid. Moreover, further details on the reconstruction algorithm are included, and a list of supplementary figures and movies.

AUTHOR CONTRIBUTIONS

WD, MRM, and SM conceived the general idea of the work. MKL developed the acoustic manipulation strategy, designed and fabricated the acoustic chamber. MKL and SD carried out the acoustic-OCT experiments at the Medical University of Vienna. SD, RL, and WD planned the OCM experiments, SM and RL developed the numerical reconstruction algorithm, and all authors contributed to the structuring and writing of the paper.

-
- [1] M. E. Sakalem, M. T. De Sibio, F. A. d. S. da Costa, and M. de Oliveira, Historical evolution of spheroids and organoids, and possibilities of use in life sciences and medicine, *Biotechnology Journal* **16**, 2000463 (2021).
- [2] S. Gunti, A. T. Hoke, K. P. Vu, and N. R. London Jr, Organoid and spheroid tumor models: Techniques and applications, *Cancers* **13**, 874 (2021).
- [3] M. T. Kozlowski, C. J. Crook, and H. T. Ku, Towards organoid culture without matrigel, *Communications biology* **4**, 1387 (2021).
- [4] E. A. Aisenbrey and W. L. Murphy, Synthetic alternatives to matrigel, *Nature Reviews Materials* **5**, 539 (2020).
- [5] M. Habaza, B. Gilboa, Y. Roichman, and N. T. Shaked, Tomographic phase microscopy with 180 rotation of live cells in suspension by holographic optical tweezers, *Optics letters* **40**, 1881 (2015).
- [6] J. Sun, N. Koukourakis, J. Guck, and J. W. Czarske, Rapid computational cell-rotation around arbitrary axes in 3d with multi-core fiber, *Biomedical Optics Express* **12**, 3423 (2021).
- [7] K. Dholakia, B. W. Drinkwater, and M. Ritsch-Marte, Comparing acoustic and optical forces for biomedical research, *Nature Reviews Physics* , 1 (2020).
- [8] G. Thalhammer, R. Steiger, S. Bernet, and M. Ritsch-Marte, Optical macro-tweezers: trapping of highly motile micro-organisms, *Journal of Optics* **13**, 044024 (2011).
- [9] Note that the limits of optical trapping stated in [12] are incorrect, significantly overestimating heating under normal operating conditions.
- [10] G. Thalhammer, R. Steiger, M. Meinschad, M. Hill, S. Bernet, and M. Ritsch-Marte, Combined acoustic and optical trapping, *Biomedical Optics Express* **2**, 2859 (2011).
- [11] M. K. Løvmo, B. Pressl, G. Thalhammer, and M. Ritsch-Marte, Controlled orientation and sustained rotation of biological samples in a sono-optical microfluidic device, *Lab on a Chip* **21**, 1563 (2021).
- [12] Z. Yang, K. L. Cole, Y. Qiu, I. M. Somorjai, P. Wijesinghe, J. Nylk, S. Cochran, G. C. Spalding, D. A. Lyons, and K. Dholakia, Light sheet microscopy with acoustic sample confinement, *Nature communications* **10**, 669 (2019).
- [13] T. Schwarz, P. Hahn, G. Petit-Pierre, and J. Dual, Rotation of fibers and other non-spherical particles by the acoustic radiation torque, *Microfluidics and Nanofluidics* **18**, 65 (2015).
- [14] A. Lamprecht, T. Schwarz, J. Wang, and J. Dual, Viscous torque on spherical micro particles in two orthogonal acoustic standing wave fields, *The Journal of the Acoustical Society of America* **138**, 23 (2015).
- [15] A. Marzo, S. A. Seah, B. W. Drinkwater, D. R. Sahoo, B. Long, and S. Subramanian, Holographic acoustic elements for manipulation of levitated objects, *Nature communications* **6**, 8661 (2015).
- [16] D. Baresch, J.-L. Thomas, and R. Marchiano, Orbital angular momentum transfer to stably trapped elastic particles in acoustical vortex beams, *Physical review letters* **121**, 074301 (2018).
- [17] M. K. Løvmo, S. Moser, G. Thalhammer-Thurner, and M. Ritsch-Marte, Acoustofluidic trapping device for high-na multi-angle imaging, *Frontiers in Physics* **10**, 940115 (2022).
- [18] C. Chen, Y. Gu, J. Philippe, P. Zhang, H. Bachman, J. Zhang, J. Mai, J. Rufo, J. F. Rawls, E. E. Davis, *et al.*, Acoustofluidic rotational tweezing enables high-speed contactless morphological phenotyping of zebrafish lar-

- vae, *Nature communications* **12**, 1118 (2021). 1056
- [19] Z. Zhang, L. K. Allegrini, N. Yanagisawa, Y. Deng, S. C. Neuhauss, and D. Ahmed, Sonorotor: An acoustic rotational robotic platform for zebrafish embryos and larvae, *IEEE Robotics and Automation Letters* **8**, 2598 (2023). 1057-1058-1059-1060
- [20] D. Ahmed, A. Ozcelik, N. Bojanala, N. Nama, A. Upadhyay, Y. Chen, W. Hanna-Rose, and T. J. Huang, Rotational manipulation of single cells and organisms using acoustic waves, *Nature communications* **7**, 11085 (2016). 1061-1062-1063-1064
- [21] N. F. Läubli, J. T. Burri, J. Marquard, H. Vogler, G. Mosca, N. Vertti-Quintero, N. Shamsudhin, A. DeMello, U. Grossniklaus, D. Ahmed, *et al.*, 3d mechanical characterization of single cells and small organisms using acoustic manipulation and force microscopy, *Nature communications* **12**, 2583 (2021). 1065-1066-1067-1068-1069-1070
- [22] A. Ozcelik, N. Nama, P.-H. Huang, M. Kaynak, M. R. McReynolds, W. Hanna-Rose, and T. J. Huang, Acoustofluidic rotational manipulation of cells and organisms using oscillating solid structures, *Small (Weinheim an der Bergstrasse, Germany)* **12**, 5120 (2016). 1071-1072-1073-1074-1075
- [23] N. F. Laubli, M. S. Gerlt, A. Wuthrich, R. T. Lewis, N. Shamsudhin, U. Kutay, D. Ahmed, J. Dual, and B. J. Nelson, Embedded microbubbles for acoustic manipulation of single cells and microfluidic applications, *Analytical Chemistry* **93**, 9760 (2021). 1076-1077-1078-1079-1080
- [24] D. Huang, E. A. Swanson, C. P. Lin, J. S. Schuman, W. G. Stinson, W. Chang, M. R. Hee, T. Flotte, K. Gregory, C. A. Puliafito, *et al.*, Optical coherence tomography, *science* **254**, 1178 (1991). 1081-1082-1083-1084
- [25] A. F. Fercher, W. Drexler, C. K. Hitzenberger, and T. Lasser, Optical coherence tomography-principles and applications, *Reports on progress in physics* **66**, 239 (2003). 1085-1086-1087-1088-1089
- [26] R. A. Leitgeb and B. Baumann, Multimodal optical medical imaging concepts based on optical coherence tomography, *Frontiers in physics* **6**, 114 (2018). 1090-1091
- [27] A. Lichtenegger, B. Baumann, and Y. Yasuno, Optical coherence tomography is a promising tool for zebrafish-based research—a review, *Bioengineering* **10**, 5 (2022). 1092-1093-1094
- [28] R. Raghunathan, M. Singh, M. E. Dickinson, and K. V. Larin, Optical coherence tomography for embryonic imaging: a review, *Journal of biomedical optics* **21**, 050902 (2016). 1095-1096-1097-1098
- [29] J. Wang, Y. Xu, and S. A. Boppart, Review of optical coherence tomography in oncology, *Journal of biomedical optics* **22**, 121711 (2017). 1099-1100-1101
- [30] A. W. Browne, C. Arnesano, N. Harutyunyan, T. Khuu, J. C. Martinez, H. A. Pollack, D. S. Koos, T. C. Lee, S. E. Fraser, R. A. Moats, *et al.*, Structural and functional characterization of human stem-cell-derived retinal organoids by live imaging, *Investigative ophthalmology & visual science* **58**, 3311 (2017). 1102-1103-1104-1105-1106-1107
- [31] A. J. Deloria, S. Haider, B. Dietrich, V. Kunihs, S. Oberhofer, M. Knöfler, R. Leitgeb, M. Liu, W. Drexler, and R. Haindl, Ultra-high-resolution 3d optical coherence tomography reveals inner structures of human placenta-derived trophoblast organoids, *IEEE Transactions on Biomedical Engineering* **68**, 2368 (2020). 1108-1109-1110-1111-1112-1113
- [32] Y. Huang, S. Wang, Q. Guo, S. Kessel, I. Rubinoff, L. L.-Y. Chan, P. Li, Y. Liu, J. Qiu, and C. Zhou, Optical coherence tomography detects necrotic regions and volumetrically quantifies multicellular tumor spheroids, *Cancer research* **77**, 6011 (2017). 1114-1115-1116-1117-1118-1119
- [33] Y. Sung, W. Choi, C. Fang-Yen, K. Badizadegan, R. R. Dasari, and M. S. Feld, Optical diffraction tomography for high resolution live cell imaging, *Optics express* **17**, 266 (2009). 1120-1121
- [34] P. Müller, M. Schürmann, and J. Guck, The theory of diffraction tomography, arXiv:1507.00466 (2015). 1122-1123
- [35] M. J. Girard, N. G. Strouthidis, C. R. Ethier, and J. M. Mari, Shadow removal and contrast enhancement in optical coherence tomography images of the human optic nerve head, *Investigative ophthalmology & visual science* **52**, 7738 (2011). 1124-1125
- [36] H. Cheong, S. K. Devalla, T. H. Pham, L. Zhang, T. A. Tun, X. Wang, S. Perera, L. Schmetterer, T. Aung, C. Boote, *et al.*, Deshadowgan: a deep learning approach to remove shadows from optical coherence tomography images, *Translational Vision Science & Technology* **9**, 23 (2020). 1126-1127
- [37] H. Cheong, S. K. Devalla, T. Chuangsuwanich, T. A. Tun, X. Wang, T. Aung, L. Schmetterer, M. L. Buist, C. Boote, A. H. Thiéry, *et al.*, Oct-gan: single step shadow and noise removal from optical coherence tomography images of the human optic nerve head, *Biomedical optics express* **12**, 1482 (2021). 1128-1129
- [38] K. C. Zhou, R. P. McNabb, R. Qian, S. Degan, A.-H. Dhalla, S. Farsiu, and J. A. Izatt, Computational 3d microscopy with optical coherence refraction tomography, *Optica* **9**, 593 (2022). 1130-1131
- [39] F. Mitri, Acoustic radiation force acting on elastic and viscoelastic spherical shells placed in a plane standing wave field, *Ultrasonics* **43**, 681 (2005). 1132-1133
- [40] H. Bruus, Acoustofluidics 7: The acoustic radiation force on small particles, *Lab on a Chip* **12**, 1014 (2012). 1134-1135
- [41] P. Glynne-Jones, P. P. Mishra, R. J. Boltryk, and M. Hill, Efficient finite element modeling of radiation forces on elastic particles of arbitrary size and geometry, *The Journal of the Acoustical Society of America* **133**, 1885 (2013). 1136-1137
- [42] G. T. Silva, J. H. Lopes, J. P. Leão-Neto, M. K. Nichols, and B. W. Drinkwater, Particle patterning by ultrasonic standing waves in a rectangular cavity, *Physical Review Applied* **11**, 054044 (2019). 1138-1139
- [43] Z. Fan, D. Mei, K. Yang, and Z. Chen, Acoustic radiation torque on an irregularly shaped scatterer in an arbitrary sound field, *The Journal of the Acoustical Society of America* **124**, 2727 (2008). 1140-1141
- [44] G. Silva, T. Lobo, and F. Mitri, Radiation torque produced by an arbitrary acoustic wave, *EPL (Europhysics Letters)* **97**, 54003 (2012). 1142-1143
- [45] N. Hari, P. Patel, J. Ross, K. Hicks, and F. Vanholsbeeck, Optical coherence tomography complements confocal microscopy for investigation of multicellular tumour spheroids, *Scientific Reports* **9**, 10601 (2019). 1144-1145
- [46] K. C. Zhou, R. Qian, S. Degan, S. Farsiu, and J. A. Izatt, Optical coherence refraction tomography, *Nature photonics* **13**, 794 (2019). 1146-1147
- [47] L. K. Young, M. Jarrin, C. D. Saunter, R. A. Quinlan, and J. M. Girkin, Non-invasive in vivo quantification of the developing optical properties and graded index of the embryonic eye lens using spim, *Biomed. Opt. Express* **9**, 2176 (2018). 1148-1149
- [48] K. Wang, I. Vorontsova, M. Hoshino, K. Uesugi, N. Yagi, J. E. Hall, T. F. Schilling, and B. K. Pierscionek, Optical development in the zebrafish eye lens, *The FASEB Journal* **34**, 5552 (2020). 1150-1151

- [49] J. Zhang, S. Yang, C. Chen, J. H. Hartman, P.-H. Huang, L. Wang, Z. Tian, P. Zhang, D. Faulkenberry, J. N. Meyer, *et al.*, Surface acoustic waves enable rotational manipulation of *Caenorhabditis elegans*, *Lab on a Chip* **19**, 984 (2019).
- [50] I. A. Favre-Bulle, A. B. Stilgoe, H. Rubinsztein-Dunlop, and E. K. Scott, Optical trapping of otoliths drives vestibular behaviours in larval zebrafish, *Nature communications* **8**, 630 (2017).
- [51] S. M. Woodside, B. D. Bowen, and J. M. Piret, Measurement of ultrasonic forces for particle-liquid separations, *AIChE journal* **43**, 1727 (1997).
- [52] Z. Ao, H. Cai, Z. Wu, J. Ott, H. Wang, K. Mackie, and F. Guo, Controllable fusion of human brain organoids using acoustofluidics, *Lab on a Chip* **21**, 688 (2021).
- [53] S. Price, S. Bhosle, E. Gonçalves, X. Li, D. P. McClurg, S. Barthorpe, A. Beck, C. Hall, H. Lightfoot, L. Farrow, *et al.*, A suspension technique for efficient large-scale cancer organoid culturing and perturbation screens, *Scientific Reports* **12**, 5571 (2022).
- [54] Y. Weng, S. Han, M. T. Sekyi, T. Su, A. N. Mattis, and T. T. Chang, Self-assembled matrigel-free ipsc-derived liver organoids demonstrate wide-ranging highly differentiated liver functions, *Stem Cells* **41**, 126 (2023).
- [55] K. Chen, M. Wu, F. Guo, P. Li, C. Y. Chan, Z. Mao, S. Li, L. Ren, R. Zhang, and T. J. Huang, Rapid formation of size-controllable multicellular spheroids via 3d acoustic tweezers, *Lab on a Chip* **16**, 2636 (2016).
- [56] K. Olofsson, V. Carannante, M. Ohlin, T. Frisk, K. Kushiro, M. Takai, A. Lundqvist, B. Önfelt, and M. Wiklund, Acoustic formation of multicellular tumor spheroids enabling on-chip functional and structural imaging, *Lab on a Chip* **18**, 2466 (2018).
- [57] N. Jeger-Madiot, L. Arakelian, N. Setterblad, P. Bruneval, M. Hoyos, J. Larghero, and J.-L. Aider, Self-organization and culture of mesenchymal stem cell spheroids in acoustic levitation, *Scientific reports* **11**, 8355 (2021).
- [58] H. Bruus, Acoustofluidics 10: scaling laws in acoustophoresis, *Lab on a Chip* **12**, 1578 (2012).
- [59] D. Bazou, R. Kearney, F. Mansergh, C. Bourdon, J. Farrar, and M. Wride, Gene expression analysis of mouse embryonic stem cells following levitation in an ultrasound standing wave trap, *Ultrasound in medicine & biology* **37**, 321 (2011).
- [60] M. Wiklund, Acoustofluidics 12: Biocompatibility and cell viability in microfluidic acoustic resonators, *Lab on a Chip* **12**, 2018 (2012).
- [61] T. S. Ralston, D. L. Marks, P. Scott Carney, and S. A. Boppart, Interferometric synthetic aperture microscopy, *Nature physics* **3**, 129 (2007).
- [62] A. Kumar, W. Drexler, and R. A. Leitgeb, Subaperture correlation based digital adaptive optics for full field optical coherence tomography, *Optics express* **21**, 10850 (2013).
- [63] Y.-Z. Liu, F. A. South, Y. Xu, P. S. Carney, and S. A. Boppart, Computational optical coherence tomography, *Biomedical optics express* **8**, 1549 (2017).
- [64] W. Drexler and J. G. Fujimoto, *Optical coherence tomography: technology and applications* (Springer Science & Business Media, 2008).
- [65] C. Macdonald, S. Arridge, and P. Munro, On the inverse problem in optical coherence tomography, *Scientific Reports* **13** (2023).
- [66] S. Deng, R. Haindl, E. Zhang, P. Beard, E. Scheuringer, C. Sturtzel, Q. Li, A. J. Deloria, H. Sattmann, R. A. Leitgeb, *et al.*, An optical coherence photoacoustic microscopy system using a fiber optic sensor, *APL Photonics* **6** (2021).
- [67] R. Haindl, M. Duell, S. Gloor, J. Dahdah, J. Ojeda, C. Sturtzel, S. Deng, A. J. Deloria, Q. Li, M. Liu, *et al.*, Ultra-high-resolution sd-ocm imaging with a compact polarization-aligned 840 nm broadband combined-sled source, *Biomedical Optics Express* **11**, 3395 (2020).
- [68] R. Haindl, S. Preisser, M. Andreana, W. Rohringer, C. Sturtzel, M. Distel, Z. Chen, E. Rank, B. Fischer, W. Drexler, *et al.*, Dual modality reflection mode optical coherence and photoacoustic microscopy using an akinetic sensor, *Optics Letters* **42**, 4319 (2017).
- [69] R. Haindl, W. Trasischker, B. Baumann, M. Pircher, and C. Hitzenberger, Three-beam doppler optical coherence tomography using a facet prism telescope and mems mirror for improved transversal resolution, *Journal of modern Optics* **62**, 1781 (2015).
- [70] J. Schindelin, I. Arganda-Carreras, E. Frise, V. Kaynig, M. Longair, T. Pietzsch, S. Preibisch, C. Rueden, S. Saalfeld, B. Schmid, *et al.*, Fiji: an open-source platform for biological-image analysis, *Nature methods* **9**, 676 (2012).
- [71] K. A. Vermeer, J. Mo, J. J. A. Weda, H. G. Lemij, and J. F. de Boer, Depth-resolved model-based reconstruction of attenuation coefficients in optical coherence tomography, *Biomed. Opt. Express* **5**, 322 (2014).
- [72] T. van Leeuwen, D. Faber, and M. Aalders, Measurement of the axial point spread function in scattering media using single-mode fiber-based optical coherence tomography, *IEEE Journal of Selected Topics in Quantum Electronics* **9**, 227 (2003).
- [73] J. Bradbury, R. Frostig, P. Hawkins, M. J. Johnson, C. Leary, D. Maclaurin, G. Necula, A. Paszke, J. VanderPlas, S. Wanderman-Milne, and Q. Zhang, JAX: composable transformations of Python+NumPy programs (2018).

REVIEWERS' COMMENTS:

Reviewer #1 (Remarks to the Author):

I am pleased with the revisions made by the author and recommend this paper for publication. Congratulations !

Reviewer #2 (Remarks to the Author):

I believe the authors have well adressed to all the comments from both reviewers, and have reasonnably argued when experiments we were asking were too time consuming or outside the scope of the manuscript. I have nothing to add, and the manuscript could be published as far as I am concerned. Conngratulations for this nice work!

Reviewer #2 (Remarks on code availability):

Too far from my expertise, and I unfortunately don't have time. It would be (more) useful if progressive feedbacks from the community regarding the code were frequently updated on the article webpage